# The role of neonatal kisspeptin in long-term social behavior in mammals
João R. Neves [1], Miguel Castelo-Branco [1,2,3,4] & Joana Gonçalves [1,2,3,4]

Kisspeptins (Kiss) are key regulators of the hypothalamic-pituitary-gonadal axis, influencing testosterone surges essential for brain masculinization and behavioral development in mammals. This study explored the effects of transient neonatal Kiss blockade on long-term social behaviors in Wistar rats. Newborn rats of both sexes were injected with either a Kiss antagonist or vehicle during the postnatal testosterone surge, termed "minipuberty". In adolescence and adulthood, social behaviors, hypothalamic Kiss receptor levels, and serum levels of Gonadotropin-releasing hormone (GnRH), luteinizing hormone (LH), testosterone, and follicle-stimulating hormone (FSH) were assessed. Results showed that neonatal Kiss modulates testosterone differently in males and females, influencing social communication and long-term social skills. Increased exploratory behavior was observed, with males exhibiting heightened sexual impulsiveness without anxiety changes. Altered hypothalamic-pituitary-gonadal hormone levels due to Kiss blockade may help explain some results. These findings highlight the critical role of neonatal Kiss in shaping lifelong social interactions and communication in a sex-dependent manner.

Kisspeptins (Kiss) are a class of peptides that are functionally relevant in a plethora of mechanisms, from metabolic to reproductive regulation[1–6]. The Kiss system is widespread in the central nervous system (CNS) of mammals, with a notable role in the hypothalamus[7–9]. Importantly, this system is distributed in a sex-dimorphic way, varying by region, and developmental stage in each sex[10–12].

Due to its role in regulating metabolism, Kiss has been extensively studied for treating metabolic dysfunction[13]. However, Kiss is also known to be influential in reproductive function by directly regulating the hypothalamic-pituitary-gonadal (HPG) axis and stimulating gonadotropin-releasing hormone (GnRH) neurons in the hypothalamus[14–16]. Studies in humans and rodents have shown Kiss's effects on improving or treating sexual dysfunction, ovarian malfunction, and other reproductive pathologies in puberty and adulthood[13,17–22].

Kiss has been shown to have time-specific expression levels throughout a mammal's life, with peaks at key milestones of sexual development and brain organization in a sex-dependent manner[11]. Research has extensively demonstrated Kiss's prominent role in the dimorphic development in the brain, particularly in brain masculinization/defeminization processes in males, as opposed to feminization processes in females, such as the perinatal testosterone surge and puberty onset[11,23–25]. During the critical perinatal window for sexual differentiation, Kiss and testosterone align in two surges[26]. The prenatal testosterone surge,

associated with gonad development, is independent of Kiss[26]. Conversely, the immediate postnatal testosterone surge (NTS), frequently referred to as the "minipuberty"[27], which has been related to gender-linked social development in humans[28], is Kiss-dependent[26]. Even during this perinatal period, Kiss seems to exert different functions in each peak, suggesting distinct developmental actions. In females, the role of Kiss in organizing networks that modulate specific phenotypes is still poorly understood.

Existing literature predominantly focuses on the effects of Kiss on sexual behavior in adult knockout (KO) mice, which does not consider the effects this system exerts in the short- and long-term periods of life. It is important to understand the impact of natural fluctuations of Kiss throughout life, such as its high levels in early life and at the onset of puberty and reduced levels in adulthood[11]. Despite the hypothesis that Kiss in the NTS is linked to sex-specific social development, there is no literature, to our knowledge, that demonstrates its lifelong role in sex-dependent behavioral phenotypes.

Notwithstanding the evidence of the role of the Kiss system in sexual dimorphism, the understanding of the lifelong effects of neonatal Kiss and its modulation during a period of high neuroplasticity and increased sensitivity to physiological fluctuations[29,30] is lacking. Even with considerable evidence on the impact of sexual hormones during this period in both males and females[29,31], little is known about the immediate and long-term effects of Kiss during this time frame and its influence on social interaction throughout the lifespan.

[1]Coimbra Institute for Biomedical Imaging and Translational Research (CIBIT), University of Coimbra, Coimbra, Portugal. [2]Institute of Nuclear Sciences Applied to Health (ICNAS), University of Coimbra, Coimbra, Portugal. [3]Institute of Physiology, Faculty of Medicine, University of Coimbra, Coimbra, Portugal. [4]These authors jointly supervised this work: Miguel Castelo-Branco, Joana Gonçalves. ✉e-mail: mcbranco@fmed.uc.pt; jgoncalves@icnas.uc.pt

In this study, we investigated the effect of neonatal Kiss peak temporary blockade on modulating juvenile and adult sex-specific phenotype. We examined this temporary blockade's impact using a comprehensive set of behavioral tests and used molecular assays to investigate alterations in the HPG axis in adulthood. This approach aimed to elucidate the long-lasting behavioral imprints of neonatal Kiss and shed light on possible molecular mediators of sex-dependent brain organization.

## Results

### Temporary neonatal Kiss blockade has sex-specific effects on testosterone levels

To investigate if Kiss transitory blockade, through an intracerebroventricular (icv) injection in the first hours of life, induced alterations in motor and vestibular development, we evaluated a set of developmental milestones from postnatal day 4 (P4) until P14. These tests encompassed their performance on righting reflex, negative geotaxis, locomotion, wire suspension, and nest seeking. No relevant or long-lasting alterations (Supplementary Fig. 1) were found. Differences were only observed at P4 in the righting reflex test, between males and females injected with Kiss antagonist (Kp234), which was immediately recovered on the subsequent day.

Further, to evaluate if Kp234 effectively blocks Kiss signaling and, consequently, testosterone peak, a set of animals was sacrificed 1 hour after injection, and plasma levels of testosterone were measured (Fig. 1). A significant main effect of sex ($F_{1,21} = 15.4$, $p$ value = 0.0008, $\eta_p^2 = 0.30$) and an interaction effect ($F_{1,21} = 11.78$, $p$ value = 0.0025, $\eta_p^2 = 0.23$) were found. Between males, the comparison between the vehicle (Veh) (1.320 ± 0.2324 ng/ml) and Kp234 (0.8266 ± 0.3082 ng/ml) showed a significant decrease ($q$ value = 6.20, $p$ value = 0.0014). Additionally, as expected, a significant difference was observed ($q = 7.107$ and a $p$ value = 0.0003) between males and females (0.628 ± 0.0279 ng/ml) Veh. However, no differences were verified between the sexes in the Kp234 groups. These findings demonstrated that temporary acute icv injection with Kp234 was able to block, at least in part, the minipuberty in males, reducing the sex difference of testosterone levels upon kiss blockade. In sum, neonatal Kiss signaling seems to be determinant in maintaining typical male minipuberty.

### Long-lasting social communication alterations are observed following neonatal Kiss perturbation

Ultrasonic vocalizations (USV) are potential markers of developmental changes in health and disease. As in humans, rats use USV as a form of communication and emit them in specific contexts[32]. We investigated two important time points to understand if temporary blockade of neonatal Kiss influenced normal communication upon maternal separation in early development (P6) and during the juvenile social play test (P29; Fig. 2). Importantly, due to technical constraints since the juvenile social play test is recorded in pairs, we were not able to isolate the USV of the animals, however, each rat was paired with a same-sex, age, and treatment partner.

Here we focused on the frequency properties of non-modulated frequency calls[33], and observed at P6 a treatment effect in all parameters – principal frequency (Fig. 2A; $F_{1,27} = 12.36$, $p$ value = 0.0016, $\eta_p^2 = 0.29$), peak frequency (Fig. 2C; $F_{1,27} = 12.08$, $p$ value = 0.0017, $\eta_p^2 = 0.28$), low frequency (Fig. 2E; $F_{1,27} = 13.27$, $p$ value = 0.0011, $\eta_p^2 = 0.31$) and high frequency (Fig. 2G; $F_{1,27} = 11.66$, $p$ value = 0.0020, $\eta_p^2 = 0.28$). Although no differences were observed between experimental groups of females at P6, males Kp234 exhibited a decrease in these call properties. The comparison between Veh and Kp234 males reported a consistent reduction in principal frequency (Fig. 2A; Veh: 58.865 ± 4.284 kHertz; Kp234:49.077 ± 5.513 kHertz; $q = 5.065$, $p$ value = 0.0068), in peak frequency (Fig. 2C; Veh: 58.821 ± 4.369 kHertz; Kp234: 49.186 ± 5.531 kHertz; $q = 4.968$, $p$ value = 0.0081), in low frequency (Fig. 2E; Veh: 56.433 ± 4.044 kHertz; Kp234: 46.946 ± 4.904 kHertz; $q = 5.275$, $p$ value = 0.0047) and, finally, in high frequency (Fig. 2G; Veh: 60.753 ± 4.045 kHertz; Kp234: 50.738 ± 5.965 kHertz; $q = 4.879$, $p$ value = 0.0095). Overall, isolation-induced call frequency at P6 was reduced upon Kiss blockade, mainly affecting males.

In the juvenile age, we explored differences in frequency properties in the USV emitted during the social play test. We only observed a significant interaction effect ($F_{1,14} = 9.660$, $p = 0.0077$, $\eta_p^2 = 0.36$) in high frequency, showing a more pronounced sex difference upon Kp234 treatment (Fig. 2H), as demonstrated by a significant decrease between males (78.998 ± 2.075 kHertz) and females (72.951 ± 3.265 kHertz; $q = 3.596$, $p$ value = 0.0227). These results indicated that temporary neonatal Kiss blockade results in long-lasting sex-dependent changes in social USV frequency, by accentuating the sexual dimorphism of USV emission in a social context, which likely affects their social interaction.

### Transient neonatal Kiss blockade reduces juvenile play tendencies independently of the sex

Our main goal was to assess whether transient Kiss blockade during the critical neonatal period induced alterations in the sociability of the animals. Therefore, at P29 we examined the social play behavior, as already described[34], such as the number of pounces, pins, and boxing (Fig. 3). Regarding the overall play (Fig. 3A), we observed a significant main effect of treatment ($F_{1,32} = 8.065$, $p$ value = 0.0078, $\eta_p^2 = 0.19$). Further, investigating more closely each behavior we found the same treatment effect on pounce (Fig. 3B; $F_{1,32} = 6.332$, $p$ value = 0.0171, $\eta_p^2 = 0.16$), pin (Fig. 3C; $F_{1,32} = 6.904$, $p$ value = 0.0131, $\eta_p^2 = 0.15$), and box (Fig. 3D; $F_{1,32} = 8.416$, $p$ value = 0.0067, $\eta_p^2 = 0.21$), all pointing to a reduction in Kp234-treated groups.

Additionally, we were interested in understanding if, on one hand, the USVs emitted during this social interaction were correlated with the social play activity, and on the other hand, if the characteristics of those USVs influenced social behavior. Interestingly, only in the Veh groups, a significant Spearman correlation between Average Social Play and the number of USVs was observed in male vehicle (Fig. 3E; $r = 0.971$, $p$ value = 0.0111) and female vehicle (Fig. 3E; $r = 0.6122$, $p$ value = 0.0261). Regarding the correlation between the Average Social Play and the Highest Frequency, only male vehicles presented a significant correlation (Fig. 3F; $r = 0.667$, $p$ value = 0.0414). We were able to disclose that although no alterations were found in the number of calls (Supplementary Fig. 2), they seem to lose their social function upon temporary neonatal Kiss blockade. We also provided evidence that the frequency of these calls impacts the sociability of the animals, mainly males.

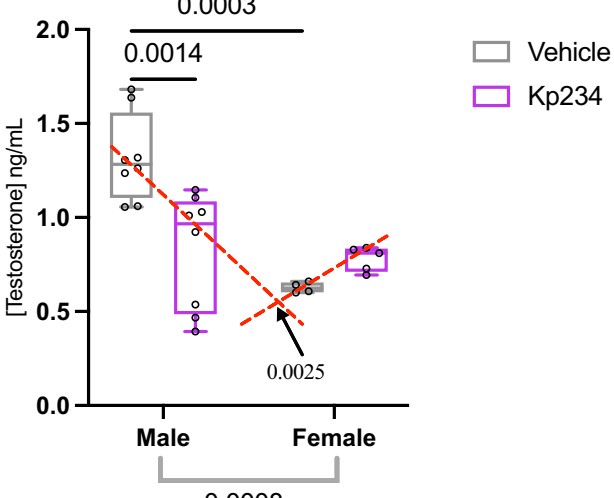

**Fig. 1 | Plasma testosterone levels 1 hour after injection evidence the importance of Kiss in mediating minipuberty in both sexes through testosterone regulation.** Two-way ANOVA, followed by Tukey's multiple comparisons test. Sample size: males Veh = 8; males Kp234 = 8; females Veh = 4; females Kp234 = 5. Data represented as box-plot min to max; circles represent individuals; numbers above the bar indicate $p$ values; red lines represent interaction effect, and gray lines represent sex effect with their respective $p$ values.

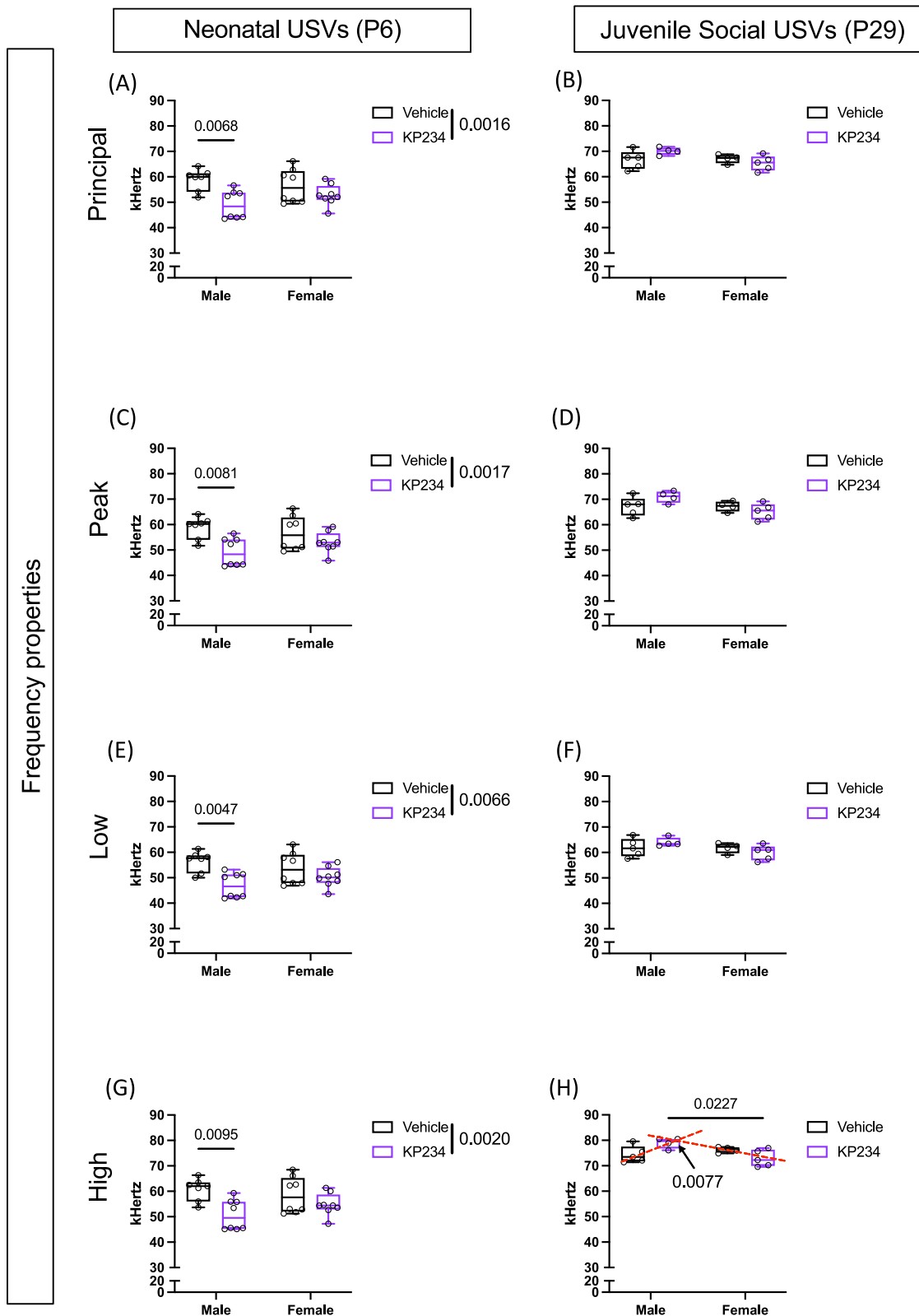

**Fig. 2 | Temporary neonatal Kiss altered early and juvenile social vocalization frequencies, especially in males.** **A** Principal frequency (median of the contour) at P6. **B** Principal frequency (median of contour) at P29. **C** Peak frequency (frequency at the highest power) at P6. **D** Peak frequency (frequency at the highest power) at P29. **E** Low frequency at P6. **F** Low frequency at P29. **G** High Frequency at P6. **H** High Frequency at P29. Two-way ANOVA, followed by Tukey's multiple comparisons test. Sample size: at PND6: males Veh = 7; males Kp234 = 8; females Veh = 8; females Kp234 = 8; at PND29: males Veh pairs = 5; males Kp234 pairs = 4; females Veh pairs = 5; females Kp234 pairs = 4. Data represented as box-plot min to máx.; circles represent individuals/pairs; numbers above the bar indicate *p* values; red lines represent interaction effect, and bars by the label represent treatment effect with their respective *p* value.

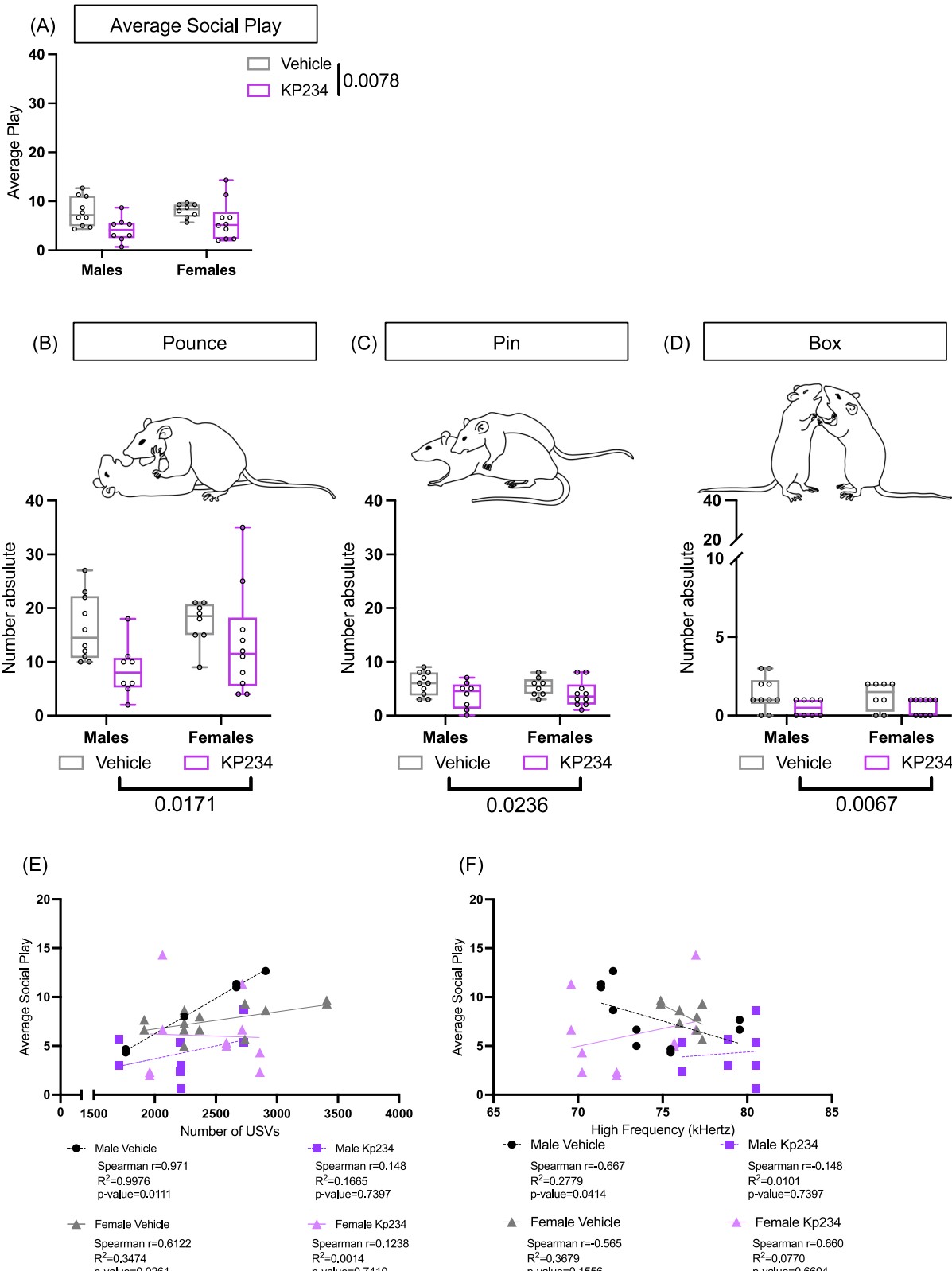

**Fig. 3 | Temporary neonatal Kiss blockade reduces social play tendencies and induces differences in the vocalization with social purpose at P29. A** Average play. **B** Number of pounces. **C** Number of pins. **D** Number of boxing **E** Correlation between average play behavior and number of USVs (for each animal, the number of calls of the pair was assumed). **F** Correlation between average play behavior and mean high frequency of USVs (for each animal, the high frequency of the pair was assumed for each). Two-way ANOVA, followed by Tukey's multiple comparisons test. The Spearman test was used to look at the correlational data. Sample size: males Veh = 10; males Kp234 = 8; females Veh = 8; females Kp234 = 10. Data represented as box-plot min to máx.; circles represent individuals; bars by the label represent treatment effect with the respective *p* value. Representative figures of behaviors were drawn manually using Adobe Illustrator v23.0.1.

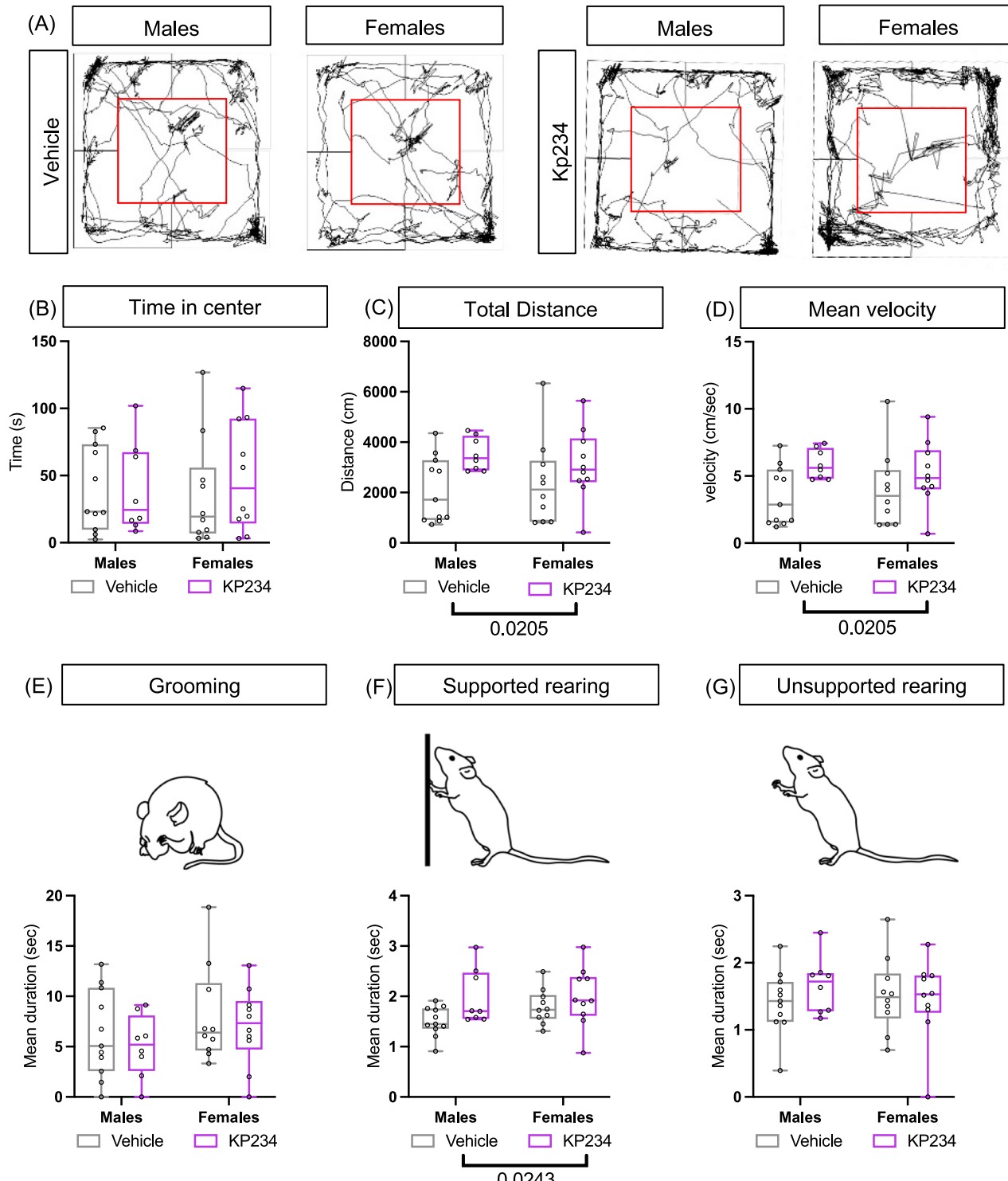

**Fig. 4 | Transient kiss perturbation in early life does not induce an anxiety-like behavior during the juvenile period. A** Representative trajectory of each experimental group throughout the test. **B** Time in the center. **C** Mean duration of grooming. **D** Mean duration of supported rearing. **E** Total distance. **F** Mean velocity. **G** Mean duration of unsupported rearing. Two-way ANOVA, followed by Tukey's multiple comparisons test. Sample size: males Veh = 11; males Kp234 = 8; females Veh = 11; females Kp234 = 11. Data represented as box-plot min to máx.; circles represent individuals; bars by the label represent treatment effect with the respective *p* value. Representative figures of behaviors drawn manually using Adobe Illustrator v23.0.1.

## Transient neonatal Kiss blockade increases exploratory activity

There is conflicting literature regarding the effects of Kiss on anxiety-like behavior[18,21,35–37]. Hence, we explored if a neonatal acute blockade of this receptor induced alterations in this behavior. At P24, we performed the open field test (Fig. 4), and a representative scheme of the trajectory is presented in Fig. 4A. We did not detect differences between the experimental groups at metrics associated with anxiety, such as time in the center (Fig. 4B), mean duration of grooming (Fig. 4E), and mean duration of supported rearing

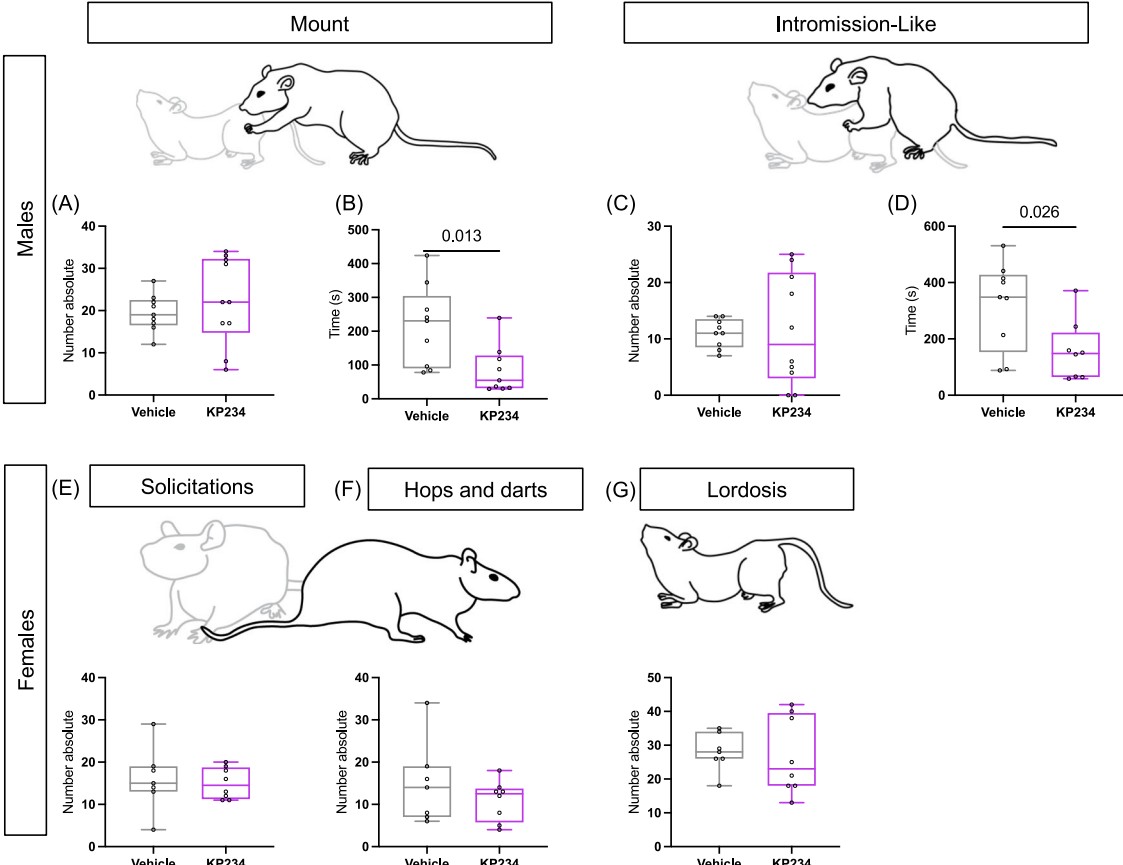

**Fig. 5 | Kiss influences masculine sexual drive during sexual behavior test in adulthood.** Male sexual behavior from (**A**) until (**D**): **A** number of mounts. **B** Latency to perform the first mount. **C** Number of intromission-like behaviors. **D** Latency to engage in the first intromission-like behavior. Female sexual behavior from **E** until **F**: **E** Number of solicitations. **F** Number of hops and darts. **G** Number of lordosis. Sample size: males Veh = 8; males Kp234 = 9; females Veh = 7; females Kp234 = 8. Data represented as box-plot min to máx.; circles represent individuals; numbers above the bar indicate significant $p$ values. Representative figures of behaviors were drawn manually using Adobe Illustrator v23.0.1.

(Fig. 4F). However, we observed for total distance (Fig. 4B) a significant treatment effect ($F_{1,35} = 5.89$, $p$ value = 0.0205, $\eta_P^2 = 0.14$), as well as in mean velocity (Fig. 4D) ($F_{1,35} = 5.89$, $p$ value = 0.0205, $\eta_P^2 = 0.14$), and unsupported rearing ($F_{1,35} = 5.5$, $p$ value = 0.0243, $\eta_P^2 = 0.13$) (Fig. 4G) which is correlated with the distance traveled[38]. Here we uncovered that Kiss, specifically in the neonatal period, does not affect anxiety-like behavior later in life, however, it seems to increase exploratory activity.

**Pre-copulatory behavior in males is altered following temporary neonatal Kiss blockade**

After P60, when animals reached sexual maturity, sexual behavior was assessed. In male sexual behavior, no differences between experimental groups were observed in copulatory behaviors, such as the number of mounts and intromission-like behavior (Fig. 5A, C). However, precopulatory behavior changes were found, such as in the latency to start both behaviors (Fig. 5B, D). Indeed, male Kp234 mount latency ($85.18 \pm 23.55$ sec) is significantly decreased compared to male Veh ($214.9 \pm 40.07$ sec; $t_{1,15} = 2.791$, $p$ value = 0.0131, $\eta_P^2 = 0.39$). The same was observed in intromission-like behavior latency, where male Kp234 ($157.4 \pm 37.85$ sec) showed a significant reduction compared to male Veh ($319.4 \pm 51.71$ sec; $t_{1,15} = 2.473$, $p$ value = 0.0259, $\eta_P^2 = 0.31$). Concerning hops and darts, solicitations, and lordosis in female sexual behavior, no statistical differences between experimental groups were detected (Fig. 5E–G). Our results revealed that although neonatal Kiss has no impact on normal sexual behavior, it impacted male precopulatory behavior, and, consequently, may have impacted male sexual drive.

**Temporary neonatal perturbation of the Kiss peak induces long-lasting changes in the HPG axis**

To understand if a transient neonatal modulation of the Kiss pathway had a long-term influence on the HPG axis, we explored the serum levels of GnRH, luteinizing hormone (LH), testosterone, and follicle-stimulating hormone (FSH) (Fig. 6A–E) after in vivo studies (P60+) and hypothalamic protein levels of Kiss receptor (KissR) (Fig. 6F) (representative unaltered and uncropped gels can be found in supplementary Fig. 3).

The levels of serum GnRH revealed a significant effect of sex ($F_{1,32} = 6.00$, $p$ value = 0.0200, $\eta_P^2 = 0.13$), treatment ($F_{1,32} = 430$, $p$ value = 0.0462, $\eta_P^2 = 0.09$), and an interaction effect ($F_{1,32} = 5.07$, $p$ value = 0.0313, $\eta_P^2 = 0.11$). These changes were mainly due to the significant difference ($q = 4.809$, $p$ value = 0.0094) between males Kp234 ($5.385 \pm 2.241$ pg/ml) and females Kp234 ($5.240 \pm 2.788$ ng/ml), and additionally the increase ($q = 4.233$, $p$ value = 0.0258) between males Veh ($6.347 \pm 2710$ pg/ml) and Kp234 ($5.385 \pm 2.241$ pg/ml). LH levels presented the same trends, with a significant sex effect ($F_{1,33} = 8.3$, $p$ value = 0.0067, $\eta_P^2 = 0.18$) and interaction effect ($F_{1,33} = 5.7$, $p$ value = 0.0226, $\eta_P^2 = 0.12$). Again, motivated by a difference ($q = 5.339$, $p$ value = 0.0034) between males ($2626 \pm 617$ ng/ml) and females ($1696 \pm 336.7$ ng/ml) injected with Kp234. Finally, testosterone levels followed the same trend, having a significant main effect of sex ($F_{1,30} = 12.35$, $p$ value = 0.0014, $\eta_P^2 = 0.24$) and an interaction effect ($F = 5,841$, $p$ value = 0.0219, $\eta_P^2 = 0.12$). These changes were mainly due to the significant difference ($q = 6.557$, $p$ value = 0.0004) between males Kp234 ($2097.654 \pm 763.620$ ng/ml) and females Kp234 ($848.303 \pm 671.320$ ng/ml), and additionally the increase ($q = 3.858$, $p$ value = 0.0490) between males Veh ($1302.903 \pm 393.131$ ng/ml) and Kp234 ($2097.654 \pm 763.620$ ng/ml).

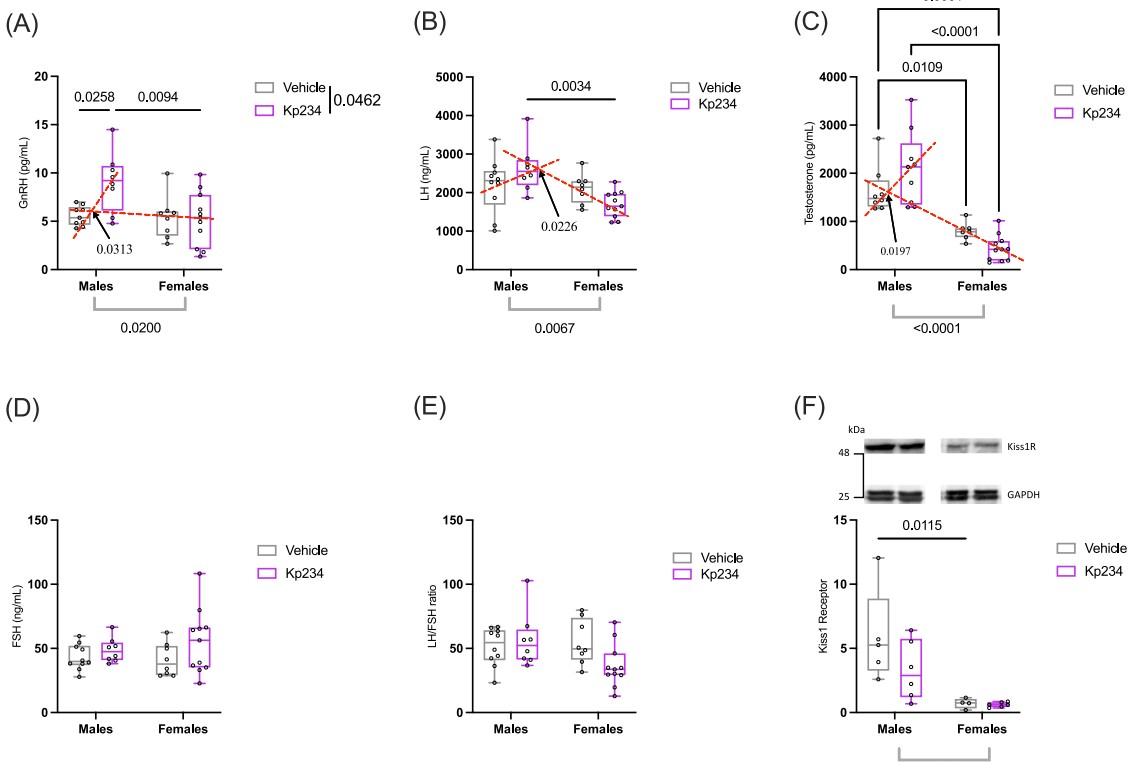

**Fig. 6 | Transient Kiss blockade during the neonatal window induces long-term changes in adult levels of hypothalamic KissR impacting HPG axis. A** Plasma gonadotropin-releasing hormone (GnRH) levels through ELISA. **B** Levels of luteinizing hormone levels through ELISA. **C** Plasma testosterone levels measured through ELISA. **D** Levels of follicle-stimulating hormone (FSH) through ELISA. **E** Ratio between LH and FSH levels. **F** KissR levels at the hypothalamic level. Two-way ANOVA, followed by Tukey's multiple comparisons test; data represented as box-plot min to max.; circles represent individuals; numbers above the bar indicate *p* values; red lines represent interaction effect and gray lines represent sex effect with their respective *p* value. Sample size: males Veh = 5–11; males Kp234 = 6–9; females Veh = 4–8; females Kp234 = 6−11.

Then we looked at the levels of FSH and the ratio LH/FSH, which are described in the literature as predictors of reproductive function[39], but we found no differences. Finally, we found a significant main effect of sex ($F_{1,17} = 16.78$, *p* value = 0.008, $\eta_p^2 = 0.44$) on protein levels of KissR (Fig. 6F). The Veh groups exhibited a significant difference between males (5.909 ± 3.643 a.u. (arbitrary units)) and females (0.713 ± 0.396 a.u.; *q* = 5.042, *p* value = 0.0115).

Overall, our data indicated that the transient neonatal Kiss blockade had a stronger long-term impact on male KissR levels. Additionally, neonatal Kiss blockade seems to have increased the sex difference in these hormones while preserving reproductive function.

## Discussion

Our study uncovered an important function of Kiss pathway during minipuberty, a critical window for brain sexual development. Our findings demonstrate that temporary disruption of this physiological process in the first hours of life induces long-lasting effects on various behaviors by reducing the gap between males and females, mainly due to loss of male sex-specific behavior, while preserving an otherwise healthy phenotype. Although Kiss has been extensively studied concerning metabolism and reproductive maintenance, our work now highlights a pivotal role in mediating social behavior and reinforces the idea that this neuropeptide exerts important general functions depending on the different stages of life.

One of the first insights supporting the complex functional profile of Kiss was the discovery that its levels fluctuate in a sex-dependent and brain region-dependent manner[40]. Although different peptide levels do not directly translate into different functions, subsequent research has explored the time-dependent functions of Kiss. It was reported that the Kiss pathway regulates testosterone levels only during the minipuberty period, while they

are regulated independently of Kiss in the prenatal period[26]. Additionally, it is worth noting that plasma testosterone levels are unaltered in neonatal KissR KO males, implying that in these genetic models, extra factors compensate for the lack of Kiss, which masks the real impact during this critical neonatal period[41,42]. Building on this, our findings align with the notion that blocking neonatal Kiss reduces systemic testosterone levels in males. Given our findings, we propose that the observed sex differences may stem from distinct neuroendocrine pathways activated by neonatal Kiss signaling. In males, blocking Kiss reduces systemic testosterone levels, likely disrupting the normal neonatal testosterone surge that contributes to brain masculinization. In contrast, our findings seem to point to a tendency of females to have an increase in the testosterone levels upon antagonism of Kiss1r. This may imply, to an extent, that Kiss in this neonatal period serves different purposes depending on the sex. However, further research is warranted to delineate the precise mechanisms and neural circuits involved in these divergent effects.

To guarantee that the changes observed in the levels of testosterone and on subsequent behavioral outcomes were exclusive of the antagonist action, it was important to observe if the icv injection by itself had no impact on motor and vestibular neurodevelopment, as these are good indicators of pediatric neurological health[43]. Our findings were consistent with existing literature, indicating no compromising effects on these outcomes by the procedure[44].

In line with humans, infant USVs in rodents are used to communicate upon a stressor presentation. During this key period, USVs are tightly aligned with physiological variations, much like a survival mechanism[33]. Therefore, during a maternal isolation protocol, pups are subjected to a highly stressful environment, both by dam separation and temperature changes between the nest and the protocol set-up[33]. Additionally, studies

have shown that Kiss neurons within the hypothalamus play a role in regulating body temperature and integrating sensory stimuli crucial for triggering appropriate behavioral responses[37,45]. Our data showed that temporary neonatal Kiss blockade, mainly in males, reduced the frequency of the calls. These changes in the neonatal period support the hypothesis that Kiss is crucial to maintaining a normal response to aversive stimuli, probably caused by temperature fluctuations in early life. However, both sexes showed the same reduction tendency in this USVs at this timepoint, which may indicate that, although males are more affected, Kiss acts irrespective of sex.

On the other hand, at the juvenile period in a social context, USVs appear to be sex-specific, and a neonatal Kiss blockade triggers a greater dimorphism, mainly motivated by an increase in male frequency and a small reduction tendency in females. A possible hypothesis for the observed changes in call frequency arises from the neurophysiological changes Kiss could induce on the neuronal circuitry of communication. Evidence in humans shows that sexual hormones, mainly testosterone, influence the frequency of the voice[46]. Also, studies in mice demonstrate that the call frequency in both males and females was higher when testosterone was administered in the neonatal and peripubertal phase, while no changes were observed when administered in adulthood[47]. The work of Kikusui and colleagues postulates that these acoustic changes are driven by a change in the main olfactory bulb-anterior olfactory nucleus-medial amygdala-hypothalamus neuronal circuitry[46]. Importantly, Kiss neurons were already hypothesized to be highly expressed and important within this circuit[21], meaning that possibly the transient neonatal Kiss blockade influenced this circuitry, causing changes in the frequency modulation of the calls in different behavioral settings and life stages. Another important finding of our work was the reduced juvenile social behavior in both sexes of Kp234-injected animals, aligning with Vanryzin and colleagues' study on the impact of testosterone on juvenile social behaviors[48]. Notably, we found important USV disconnections between the number of calls and social behaviors in Kp234-injected groups, and more importantly, we uncovered a negative correlation between the mean high frequency and the average social play, specifically in males. Therefore, a neonatal transient blockade of Kiss could change the way male rats engage in social behavior by modulating USV frequency, which compromises their social purpose. Although some authors suggest no effect of USVs on social response[49], consistent literature considers USVs pivotal for proper social interaction[33,50–54]. The relationship between USVs and Kiss is poorly documented, though a study by Asaba and colleagues showed that male USVs activate Kiss neurons in the arcuate nucleus of the hypothalamus (arc), increasing approach behavior[17]. This finding adds depth to the hypothesis that this transient Kiss blockade affects the main olfactory bulb-anterior olfactory nucleus-medial amygdala-hypothalamus neuronal circuitry, which leads to changes in the frequency of the calls and consequently affects normal social behavior. However, more studies are required to better understand the relationship between neonatal Kiss and the build-up of the neuronal circuitry governing USV modulation and social interactions.

Many studies investigate Kiss's therapeutic potential for anxiety and depressive-like behaviors. Although some literature reports contrary effects, most demonstrate, both in rodents and humans, that Kiss agonists have an anxiolytic and antidepressant effect when administered in adulthood[18,35–37]. Therefore, we might speculate that a neonatal Kiss blockade induced an increased anxiety-like behavior throughout life. However, our work showed that transient neonatal Kiss does not induce a juvenile anxiety-like behavior, indicating that in this time frame, Kiss does not intervene, at least directly, in the neuronal mechanisms underlying anxious responses. Instead, we found that Kiss blockade increased exploratory behavior in the open field test, irrespective of sex, since both males and females showed the same tendency. Additionally, males showed a reduction in pre-copulatory behavior. These findings led us to hypothesize an effect of neonatal Kiss on neurobehavioral inhibition. Previous studies have shown that Kiss administration enhances limbic and paralimbic brain activity with correlations to reward, sexual aversion, and positive mood in humans, reduction of fear responses,

elicitation of erections, and increase of pre-copulatory behavior, in other mammals[55,56]. All these outcomes share the premise of the involvement of limbic, sexual drive, and reward networks and a possible role for neural inhibition. Further, Kiss neurons have also been described as regulators of the inhibitory response through gamma-aminobutyric acid (GABA)ergic regulation[57]. We hypothesize that transient neonatal disruption of the Kiss system induces an alteration in the Kiss input in GABA neurons, promoting a reduction in inhibitory behavior and consequent increased impulsiveness to engage with the environment and other individuals. However, a piece of literature also reports high levels of KissR in key locomotor regulatory centers such as the striatum and amygdala[7,8,58]. Since a widespread icv was performed, it is plausible that neonatal Kiss antagonism promoted a different rewiring of these regions directly.

Additionally, while it may appear contradictory that neonatal Kiss blockade reduces sex-related behaviors yet is associated with increased sexual motivation in adulthood, we propose that these seemingly inconsistent findings reflect the engagement of distinct neural circuits. Specifically, we hypothesize those pre-copulatory behaviors, which are more closely linked to motivation and impulsivity, are governed by different pathways than those controlling consummatory sexual behaviors (such as mounting and intromission) in both males and females[59]. Thus, the neonatal injection may have altered the latency to engage in sexual behaviors by modulating circuits related to motivational drive, without fundamentally disrupting the underlying mechanisms of sexual performance. This interpretation suggests a differentiated modulation of neuronal circuitry, where the transient neonatal Kiss blockade impacts various aspects of sexual behavior through separate, functionally distinct networks. Therefore, more studies are required to better understand how Kiss in this short and precise developmental stage modulates the neuronal networks that govern such behaviors. Also, we did not find differences in female sexual behavior, we hypothesize is due to the metrics we looked at being more related with consummatory behaviors rather than motivation and impulse, we recognize that new behavioral paradigms should be conducted in the future to access female sexual motivation, such as using bil-eveled chambers or odor preference.

Most importantly, we examined the long-term effects of temporary neonatal Kiss perturbation on the HPG axis. Sexual dimorphism in the Kiss system fluctuates with developmental stage and region, even within the hypothalamic nuclei, becoming more pronounced later in life, particularly in the anteroventral periventricular nucleus and arc[60]. Despite strong evidence of dimorphic Kiss neurons within the hypothalamus[11,60], there is no confirmatory evidence on the levels of this receptor in this brain region. Our results demonstrated increased Kiss1r protein levels in males compared to females, with a significant decrease in males upon neonatal perturbation, at odds with the evidence regarding the number of Kiss neurons[12]. Importantly, due to space constraints, we ran the western blots in a separate membrane for each sex, but performed the experiment simultaneously, although it may increase variability between sexes, the effect size of the statistical analysis points to a strong sex effect on the receptor levels. These findings suggested that transient neonatal Kiss blockade induces long-term effects on the wiring of hypothalamic networks.

Additionally, serum levels of GnRH, LH, and testosterone exhibited patterns consistent with normal fluctuations of the HPG axis, where increased GnRH leads to elevated LH and subsequently higher testosterone levels[61]. These results indicate that neonatal Kiss perturbation modifies hormonal regulation within the HPG axis in a sex-specific manner, again, mainly explained by the increased hormone levels in males Kp234 which increased in this condition a more pronounced sex dimorphism, as evidenced by significant interaction effects, showcasing the long-term effect neonatal Kiss plays in the modulation of the HPG axis. Notably, no differences were observed in FSH levels or the LH/FSH ratio, metrics associated with spermatogenesis and ovulatory regulation[39,62]. This supports our hypothesis that neonatal Kiss signaling plays a more critical role in reprogramming hormonal systems and influencing specific behaviors, rather than directly affecting sexual performance or reproductive health.

Importantly, we observed a significant difference in testosterone levels between male and female vehicle-treated groups. This is consistent with GnRH and LH levels. Importantly, existing literature indicates that testosterone levels can be influenced by factors such as circadian rhythms, sociability, and females also experience testosterone fluctuations[63]. Our study did not account for the female estrous cycle, nor were we able to measure estradiol, which is the active form of testosterone in the brain by the action of the aromatase, which could contribute to observed variations. Future studies should consider the effect that a neonatal temporary Kiss blockade plays in the activity of aromatase, by measuring estradiol and receptor levels later in life. By doing so, we could understand if this increased sexual dimorphism arises from Kiss's direct wiring of the HPG axis or through the modulation of the conversion of testosterone into estradiol, which might be causing the changes we observed both in hormonal levels and behavior. Recent research highlights the role of testosterone pulses in females[63], suggesting that our findings may reflect genuine physiological differences that warrant further investigation. We acknowledge that more targeted studies are necessary to fully elucidate these observations.

It is important to consider several aspects and potential limitations when interpreting our findings. Firstly, this work provides valuable insights into the role of neonatal Kiss in mediating social behaviors and focuses on the HPG axis. However, it is important to remember the widespread nature of the neonatal blockade that could have influenced other mechanisms in other brain regions and, therefore, contributed to the observed phenotype. Further, recently it was demonstrated that astrocytes not only have KissR but also that the astrocytic Kiss effect is important for HPG axis function[64]. This raises questions about whether the Kiss function is also cell-type dependent and the specific role of Kiss in each type, affecting the development of sex-specific brain structures. More studies should be conducted to understand the mechanisms behind Kiss within each cell type and their impact on behavior. Secondly, while our behavioral assays provided significant insights into social interaction and exploratory behaviors, the complexity of these behaviors warrants further detailed analysis and the inclusion of additional behavioral paradigms, such as the Y-maze, to assess cognition. Importantly, this work only scratches the surface of the effect of neonatal Kiss on behavior. A more detailed mechanistic approach is needed to understand the impacts of Kiss on the levels of other important neurotransmitters and neuropeptides, how this affects the neuronal circuitry, how these alterations remain unchanged throughout the lifespan, and how later Kiss peaks, such as those during early puberty, are dependent on or independent of the neonatal peak. Another important consideration is the reported number of Kiss neurons in cisgender and transgender individuals, where male-to-female transsexuals showed a female-typical Kiss expression[65]. Predicting an important role of this system in gender identity development[28]. Therefore, having a translational perspective on Kiss research is paramount; by using animal models and integrating the information with non-invasive techniques in humans, we could better understand how the Kiss system modulates such complex behavior.

This work highlights the importance of neonatal Kiss during the minipuberty period for the proper development of sex-specific behaviors through the hypothalamic system. By focusing on developmental timing, we demonstrated the specific effects of neonatal Kiss on contextual stimuli integration, through control of inhibition and subsequent social responses, shaping a sex-specific phenotype. Future research on Kiss should consider time-specific approaches to accurately address its physiological roles.

## Materials and methods
### Animals
Female and male Wistar rats were initially obtained from Charles River Laboratories (Germany) and housed under standard laboratory conditions at the animal facility of the Institute of Nuclear Sciences Applied to Health (ICNAS), University of Coimbra. Animals underwent an acclimatization period of at least 14 days in the facility under controlled temperature ($\pm 37\,°C$), humidity, and a 12/12 h light/dark cycle before breeding. Pregnant females were allowed to deliver naturally, with the day of birth designated as postnatal day 0 (P0). Within the first 24 h after birth, neonatal pups were randomly assigned to one of four experimental groups based on sex and treatment: Veh-treated males, Kp234-treated males, Veh-treated females, and Kp234-treated females. Each animal received a single ICV injection of either vehicle (Veh; control group) or the Kisspeptin antagonist Kp234. Control animals were injected with vehicle to account for procedural effects. A sham (uninjected) group was not included, as the primary aim was to evaluate the specific role of Kisspeptin signaling during the neonatal period.

To ensure blinding, pups were tattooed with non-identifying marks immediately after injection. Experimental units were randomly picked from the cage for treatment allocation; although no formal sequence was used, all data collection and analysis were conducted blinded to minimize bias. Group allocation was known only to the person performing the injections at the time of treatment; all subsequent stages—including experiment conduct, outcome assessment, and data analysis—were performed blinded to ensure unbiased results. To minimize litter effects, pups were evenly distributed across experimental conditions. A total of nine litters were used. The experimental unit in this study was the individual rat. Animals were housed with their dams until P21. After weaning, they were grouped by sex and maintained until adulthood. Behavioral assessments were conducted between P4 and P60+ in a dedicated quiet room with controlled lighting and temperature, as specified for each test. After behavioral testing, animals were sacrificed for blood and brain collection. No exclusion criteria were established a priori; therefore, all animals were included in the statistical analyses. However, not all individuals were subjected to every test due to technical constraints or allocation of biological material to specific endpoints. Exact group sizes are indicated in the figure legends.

To ensure ethical animal use in accordance with the 3Rs principle, sample size was calculated a priori using G*Power 3.1 (Universität Kiel, Germany), based on a fixed-effects ANOVA design (main effects and interactions), with an expected effect size of 0.6, $\alpha = 0.05$, power $= 0.95$, numerator degrees of freedom $= 3$, and four groups. The resulting estimated sample size was 59 animals (~14 per group). We have complied with all relevant ethical regulations for animal use. All experimental procedures were conducted in accordance with the European Union Council Directive (2010/63/EU), Portuguese national legislation, and institutional regulations. Ethical approval was obtained from the Portuguese authority DGAV and the ICNAS Animal Welfare Body (ORBEA; approval number 5/2021).

### Intracerebroventricular injection
In the first 24 h after birth, male and female rats underwent icv injection. One by one, animals were cyio-anesthetized. When they became bluish and presented no signs of paw reflex, animals were placed in the stereotaxic apparatus with the head between the ear bars (reversed), parallel to the x-axis, and the bregma-lambda line placed parallel to the y-axis. The syringe (32 G needle (Hamilton, 17762-05) and $10\,\mu L$ injection syringe (Hamilton, 1700 series)) was placed above lambda, and X and Y coordinates were set to zero. The stereotaxic coordinates for the bilateral injection were ($\pm 0.90$;4.2;1.70) (X; Y; Z) mm, to hit the ventricles. $1\,\mu L$ of either saline (0.9% NaCl) or Kp234 (10 nmol in 0.9% NaCl solution, Tocris Bioscience, USA) solution was slowly injected, and the needle was kept in place for around one minute. This procedure was repeated in the opposite hemisphere, bilaterally. The pups were put on a warming pad until fully recovered, then returned to their mothers and home cage, and were observed until proper mother nursing. A cohort of animals was sacrificed by decapitation one hour after injection, and their blood was collected. The timepoint, antagonist selection, and concentration were chosen based on Roseweir et al.[66].

### Developmental milestones tests
Developmental milestones were conducted at PND4, 6, 8, 10, 11, 12, and 14. These time points were selected to capture key stages of sensorimotor and neurological development in neonatal rats. Always between the afternoon period (12 pm and 6 pm), in a quiet room, under dim light with temperature and humidity controlled, and a 30-second interval was taken between each

test. The tests conducted were the righting reflex, negative geotaxis, locomotion, wire suspension and nest seeking have already been described[43,44].

## Open field test
At PND24, rats were placed in the open arena and allowed to explore freely for 10 min[44]. The floor had bedding to allow proper recording, the light was set to the minimum possible (less than 50 lux), and the test was always performed between 12 pm and 6 pm. The time, mean speed, and total distance in each zone were automatically evaluated using Smart Video Tracking Software version 3.0.06 (Panlab, Barcelona, Spain). Also, lateral recordings were used to evaluate the time animals spent on supported and unsupported rearing and grooming behaviors using Behavioral Observation Research Interactive Software (BORIS)-App v. 0.9[67]. Between each animal, only the feces were collected.

## Juvenile social play behavior test
Between PND24 and PND29, animals were isolated to enhance social behavior. The juvenile behavior was assessed on PND29. Pairs of rats (sex-, treatment-, and age-matched) were placed in an open arena and left to explore freely for 12 minutes (2 min of habituation and 10 min of effective test). The floor had bedding to make the ambiance less stressful and allow proper recording, a red light was set to the minimum possible (<50 lux), and the test was always performed between 12 am and 6 pm. Video recordings were taken from the ceiling and the side to ensure proper behavior scoring. Three behaviors were scored as previously documented[34]. Behavior scoring was performed manually using (BORIS)-App v. 0.9. USVs were recorded during the test.

## USV recordings
USVs were recorded in 2 contexts: During isolation-induced USV, at PND6 in an anechoic chamber for 5 min, and during the juvenile social play behavior test at PND29 for the length of the test. Due to technical limitations, the pair was considered a single statistical unit. USVs were captured using an ultrasound recording system that incorporated an Avisoft CM16/CMPA condenser microphone. This setup included an UltrasoundGate 416H amplifier and Avisoft Recorder software (Avisoft Bioacoustics, Glienicke/Nordbahn, Germany).

After the recording of USVs, spectrograms were produced using an FFT length of 512 points, a 16-bit format, a sampling frequency of 250 kHz, a time resolution of 1 ms, a frequency resolution of 488 Hz, and an overlap of 50%. The USVs were further assessed utilizing the MATLAB toolbox DeepSqueak version 3.1.0. This facilitated the extraction of individual rat USV fragments through the application of the Rat Call_Detector_Neural network. The analysis utilized a chunk length of 6 s, an overlap of 0.1 s, a high frequency cut-off set at 125 kHz, and no score threshold. Key parameters such as principal frequency (median), lowest frequency, highest frequency, delta frequency, and peak frequency (frequency at the highest power) of each USV were subsequently extracted.

## Sexual behavior test
At PND60, animals underwent a 20-minute sexual behavior test conducted within the dark period, one hour after the beginning of the dark phase, in a quiet room where animals were placed at least 2 h before the test for habituation. This test was conducted to assess adult reproductive behavior, reflecting long-term effects of neonatal Kisspeptin signaling disruption. These tests were recorded for later analysis using two video cameras (Life-Cam HD-3000, Microsoft, Redmond, WA, USA) positioned on the room's ceiling and in a lateral position. Intact male rats were assessed for copulatory behaviors in response to natural cycling and experienced female rats, whenever they were receptive (previously tested with a 2-minute interaction with an experienced male). The behaviors recorded included the count of mounts and intromission-like behaviors. Additionally, the latency to the first occurrence was documented. Female rats with their natural reproductive cycles intact were observed for proceptive behaviors and lordosis when exposed to experienced male rats. The observations were made daily

throughout the estrus cycle, which typically lasted for 5 days, with each observation session lasting 20 min. The observed behaviors included proceptive behaviors such as hops (brief jumps with a crouched stance), darts (short sprints followed by a crouched position), and solicitations (facing the male before moving away and adopting a presenting posture). Additionally, lordosis was assessed in response to male rats actively mounting the females. After performing this test, animals were sacrificed by decapitation, and their weight and length were measured. Also, testicles were extracted and weighed.

## ELISA
Plasma from the animals sacrificed after one hour and in animals sacrificed after PND60 was isolated by centrifugation in EDTA-coated tubes, and samples were used in the ELISA kit. The manufacturer's protocol was followed (ELISA Kit for Testosterone, ref:CEA458Ge, lot: L250128814; ELISA Kit for FSH, ref: CEA830Ra, lot: L250128839; ELISA Kit for GnRH, ref:CEA843Mi, lot:L250128831 and ELISA Kit for LH, ref:CEA441RA, lot: L250128907, all from Cloud-clone Corp., Houston, USA).

## Western blot
Rat brains were extracted, and the hypothalamus was carefully dissected on ice and subsequently stored at −80℃. The hypothalamus was homogenized using a sonicator in a radioimmunoprecipitation lysis buffer [150 mM NaCl, 50 mM tris-base, 5 mM EGTA (ethylene glycol-bis(β-aminoethyl ether)-N,N,N',N'-tetraacetic acid), 1% Triton X-100, 0.5% sodium deoxycholate, 0.1% sodium dodecyl sulfate, and pH of 7.5]. This mixture was supplemented with protease inhibitors (Roche Diagnostics GmbH, Germany) and 0.1% dithiothreitol. To quantify the sample concentrations, a portion of each sample was subjected to a bicinchoninic acid (BCA) assay (Thermo Fisher). Subsequently, all samples were standardized to a concentration of 50 μg of protein per well. After migration, proteins were transferred to PVDF membranes (Merck Millipore, Ireland), and Western blotting was performed according to standard procedures. The membranes were then incubated overnight at 4℃ with the primary antibody rabbit anti-Kiss1 (GRPR54) (ref:AKR-001, lot: AKR001AN0450, Alomone Labs, Israel) at a dilution of 1:500. Also, the membranes were exposed to an alkaline phosphatase-conjugated secondary antibody (anti-rabbit 1:10,000; Thermo Fisher, U.S.A). Densitometry analysis of the resulting bands was conducted using Bio-Rad's ImageLab software (version 6.1.0). Protein expression levels were normalized against anti-glyceraldehyde-3-phosphate dehydrogenase (anti-GAPDH).

## Statistical analysis and reproducibility
The statistical analysis was performed using GraphPad Prism version 8.0.1 (GraphPad Software, San Diego, CA, USA) at a significance level of $p < 0.05$, in the context of this study, no confounders were taken into account. Data distribution was assessed using the Shapiro-Wilk test, while Levene's test was used to test for homogeneity of variance. Normal distribution was verified, parametric tests were employed, and the results were reported as $\bar{x} = $ mean ± extremes. When comparing two groups, a two-tailed independent samples t-test was used (a Welch's correction was applied if homoscedasticity was not met). To compare more than 2 groups the results were analyzed with two-way ANOVA followed by Tukey's multiple comparisons test for sex and treatment differences, were it was reported the $q$ value (studentized range distribution (similar to $t$ value); calculated as sqrt(2)*D/SED), to assess the effect size of the significant factor differences partial eta-squared ($\eta_p^2$) was calculated has $SS_{factor}$(sum of squares of the factor)/ $SS_{total}$ (total sums of squares for all effects). To analyze correlations between variables, Spearman tests were used. Graphs were constructed on GraphPad Prism version 8.0.1 (GraphPad Software, San Diego, CA, USA).

## Reporting summary
Further information on research design is available in the Nature Portfolio Reporting Summary linked to this article.

## Data availability

All the data used in the figures are available as a table in the supplementary information. Any raw data, such as videos of behavior and images, could be given upon request.

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

## Acknowledgements
This work was supported by the FCT Exploratory Project 2022.01066.PTDC and FCT Strategic plan FCT/UIDP&B/04950/2025 (CIBIT).

## Author contributions
J.R.N.—Data curation, formal analysis, methodology, software, visualization, writing—original draft. M.C.B. and J.G.—conceptualization, funding acquisition, methodology, resources, supervision, validation, writing—review & editing.

## Competing interests
The authors declare no competing interests.
