## [Transparent Peer Review file · Communications Biology]

Neonatal Kisspeptin Blockade Modulates Sex-Dependent Social Behaviors and Communication in Wistar Rats

Corresponding Author: Dr Joana Gonçalves

Version 0:

Reviewer comments:

Reviewer #1

(Remarks to the Author)

The study by Neves JR et al. examined the involvement of neonatal kisspeptin in the regulation of sexual dimorphism by intracerebroventricular (icv) injection of kisspeptin antagonist (Kp234) into postnatal rats, and subsequent analyses of its long-term effects, such as serum testosterone levels, USVs, kisspeptin-receptor expressions, and social/sexual behaviors. The authors showed early kisspeptin blockage could affect sexual dimorphism even in the juvenile and adult stages. Each result seems clear and it is very interesting. I also appreciate that the various experiments were performed at different developmental stages. However, there are inconsistencies among some of the results and the overall scheme seems obscure. I have several concerns that need to be addressed by the authors to improve the quality of this article.

Major concerns:

- Overall, the number of examples included in most experiments is insufficient. In particular, in the experiment in Figure 1, the number of samples is $n=3-4$, and in Figure 6, there is a group with only $n=2$. It is necessary to increase the number of samples in order to improve the validity of the results. Additionally, to validate functions of neonatal kisspeptin, it will be more convincing to analyze whether the opposite effects to antagonist (Kp234) icv injection were found by kisspeptin agonist icv injection.
- In some experiments, for comparison between control vehicle (Veh) groups and Kp234 groups, statistical analyses were performed using males and females separately, while in others, using combined both sexes. However, since this work focuses on sexual dimorphism, the number of samples of both sexes should be increased and all analyses should be performed separately for males and females.
- Page 4, Lines 139-141: Data in Figure 1 show that, by Kp234 icv administration, serum testosterone levels were significantly decreased in males, while increased in females at P6 stage. This suggests that neonatal kisspeptin blockage by Kp234 has inhibitory effects on masculinization or defeminization in males while it has excitatory effects in females. What brought about this sex difference? The authors should discuss some possible hypothesis or scheme. Moreover, if there is a long-lasting kisspeptin effect on development of neuronal circuits involved in social behaviors, this result seems inconsistent with the results showing shorter latency of sexual behaviors in Kp234-injected adult males. Because Kp234 inhibited masculinization of males at neonatal stage (P6), while it increased male's motivation for sexual behavior at mature adult stage (P60). The authors need to discuss this point as well.
- Page 5, Lines 166-167: Data in Figure 2 show that Kp234 induced USVs with lower frequencies at neonatal stage (P6). It is important to examine whether this change in USV frequency functionally affects social interaction (with their mothers?). Additional analysis of maternal behaviors such as pup retrieval or pup care should be performed using Kp234 injected pups, vehicle injected pups, and their mothers.
- Page 5, Lines 173-174: Data in Figure 2 also show that, at juvenile stage (P29), Kp234 administration tended to increase USV frequency in males and conversely decrease in females and it is understandable that there was some difference between the sexes in Kp234 group. However, there is no significant difference between the Veh and Kp234 groups in both males and females. Although the authors argue that the effect of Kp234 is also present in P29, it is more reasonable to interpret these results as suggesting that Kp234 has little or no effect on USV frequency changes at P29.

· Page 5, Lines 181-185: Data in Figure 3 show that Kp234 decreased juvenile social play behaviors at P29 by comparison of Veh group and Kp234 group. Is there significant difference when it is separately tested for males and females? As the authors argue that Kp234 effect on USV frequency in P29 is different between the sexes and that there is a correlation between USV and social play, it should be tested separately for males and females.

· Pages 5-6, Lines 188-195: Similar to above, data in Figure 3 also show, only in Veh group, a positive correlation between the number of USVs and the number of social plays and a negative correlation between high frequency of USV and the number of social plays. This suggests that the individuals with higher number and lower frequency of USVs exhibit higher number of social plays. However, Kp234 appears to have almost no effect on USV frequency (Figure 2) or the number of USV (Figure S2) at P29. Nevertheless, Kp234 reduced the number of social plays. These results also seem inconsistent and this point should be addressed by the authors as well.

· Page 6, Lines 208-210: Data in Figure 4 show that Kp234 did not affect anxiety-like behaviors but it increased total distance and mean velocity in open field test. Regarding this point, the authors argue that Kp234 can increase exploratory activity. It is questionable whether an increase in travel distance and velocity is equivalent to an increase in exploratory behavior. In fact, the result of open field test showed that there was no significant difference in "Time in center" and this may suggest that Kp234 affects not exploratory activity but locomotion itself. Additional behavior experiment such as Y-maze test is necessary to examine possible changes in exploratory activities. It should be noted that the possibility of Kp234 having some psychiatric effects such as excitement or hyperactivation cannot be ruled out. Such effects could potentially result in increased travel distance and velocity.

· Page 6, Lines 224-226: Similar to above, data in Figure 5 show significantly shorter latency of sexual behaviors in adult males but no effect on female behaviors. At P6, Kp234 decreased testosterone levels in males and inhibited masculinization and probably development of neuronal circuits that are involved in male-type behaviors. Nevertheless, at P60, Kp234 males exhibited higher motivation for mating. I think this is also contradictory. Besides, it should be given a plausible explanation that testosterone levels in females increased at P6 by Kp234 (defeminization), but no effect on sexual behaviors in adults.

· Page 7, Lines 234-236: As for the data shown in Figure 6 (A), I have several concerns. First, which two groups were tested resulting in $p=0.0008$? The authors argue that Kp234 affects kisspeptin receptor (KissR) expression. However, in male (also female), there was no significant difference between Veh and Kp234 group, suggesting that Kp234 does not affect the expression level of KissR. If there is KissR reduction by Kp234 in male adults, KissR reduction should cause impairment of functions of the HPG axis regulation. In turn, this impairment should negatively affect male's sexual behavior, which is actually upregulated (Figure 5). This is also contradictory and needs to be addressed. By comparing Veh groups of male and female, there is a significant sexual dimorphism in KissR expression (male >> female). Is this difference physiologically normal? Is there previous studies showing sex difference in KissR expression levels in rodents?

· Page 7, Lines 236-244: Data in Figure 6 (B) show that, in males, Kp234 significantly increased testosterone levels at P60 (adult stage), while tended to decrease them in females. By comparing Veh males and Veh females, it is suggested that testosterone levels are not different between sexually mature males and females. This result is not convincing because testosterone levels in mature males are normally much higher than those of mature females. At P6 (neonatal stage), Kp234 decreased testosterone levels in males. In contrast, at P60 (adult stage), testosterone levels were higher in Kp234 male group than Veh male group. If early kisspeptin blockage caused inhibiting effect on masculinization, these result in P60 is also inconsistent with those of P6. The authors should discuss this point as well.

· Page 7, Lines 245-249: The authors argue that Kp234 can affect the feedback loop of the HPG axis regulation. However, these data showed only KissR and testosterone levels and, in order to confirm this Kp234 effect, it is necessary to examine expression levels of other key factors (hormones) of the HPG axis regulation, such as GnRH, LH, FSH, estrogen, and kisspeptin. Quantitative PCR (qPCR) will be the most effective analysis to examine the gene expression levels of these factors. There is no correlation between KissR expression and testosterone levels in Veh groups, suggesting that KissR is expressed regardless of testosterone levels at P60. If a positive correlation between these in Kp234 group is correct, it is not consistent with the result that testosterone level was increased (B) but KissR expression is decreased (A) in Kp234 group. It is difficult to interpret these different correlation patterns between the two groups.

· Page 7, Lines 252-266: Similar to above, the different correlations between these two groups are also difficult to interpret and the correlation heatmap in Fig. 6 (D) is very complicated and difficult to interpret. For example, how do the authors explain the result that there is a positive correlation between the frequency of USVs in juveniles (P29) and the expression levels of KissR in Adults (P60)? As they are separate indices at different developmental stages, it seems unlikely that these two indices have some clear relationship.

· Pages 8, Lines 311-318: I believe that histological analysis using Kp234 icv injected rats to examine whether neonatal kisspeptin affects developmental regulation of neural circuits that are involved in social behaviors will be direct evidence for this part of the argument. Regarding classical neurotransmitters such as GABA, which is mentioned in Page9, line360, it would be very insightful to examine potential alterations in the number of neurons or projection sites.

· Pages 10, Lines 380-381: As with the above, it is not persuasive to make this argument because there was no significant difference between the sexes in the Veh group in Figure 6 (B).

Minor concerns:

Graphical abstract: Under the list with arrows indicating the actions of Kp234 icv, the text “Neonatal Kiss induces~” is presented, which makes it difficult to tell whether this list shows the action of Kp234 or the function of endogenous kisspeptin. It would be preferable to have a heading such as “Kiss antagonist effects” above the list.

· Supporting information S1 (C): It seems that there may be a significant difference in males between PND10-12. Is this really not significant (n.s.)?

· Page 5, Line 163: it should be written as “ $q=4.968$ ”.

Reviewer #2

(Remarks to the Author)

This paper investigated the role of neonatal kisspeptin in the development of social-sexual behaviors in rats. Therefore, male and female rats received at the day of birth an injection with the kisspeptin antagonist kisspeptin-234 bilaterally in the lateral ventricle or vehicle as control. Animals were then later tested for vocalizations (at different time points, postnatal day 6 and P29 during the social play test), anxiety/locomotion, social play behavior, and sexual behavior in adulthood. In a terminal experiment, expression levels of the kisspeptin receptor Kiss1r were measured using western blot, as well as adult testosterone levels. It was found that an icv injection with kisspeptin-234 affected testosterone levels immediately after the injection, with a decrease observed in males and a slight increase in females. Ultrasonic vocalizations were mostly affected in males at P6, but not as much at P29. Social play was affected in both sexes to some extent. Furthermore, some minor effects were reported in exploratory activity and male sexual behavior (decreased latencies to the first mount/intromission). Finally, hypothalamic Kiss1R levels were decreased by neonatal kisspeptin-234 treatment in males but not females. The main issue with the paper is the over-interpretation of the data, contributing a direct role to kisspeptin to the development of social and sexual behavior. There is no discussion/mention of the role of kisspeptin in stimulating GnRH neurons and how all these effects might just simply be due to decreased GnRH signaling and decreased stimulation of gonadal sex hormones (which obviously was the case regarding testosterone levels), which has affected the process of masculinization/defeminization to some extent. There is also no data presented on the duration of these effects of blocking kisspeptin receptors. Did they actually compare different doses and timelines of their treatment? They just measured testosterone levels once, 1h after the icv treatment.

The paper is just filled with overstatements, e.g. line 250, “since temporary Kiss pathway modulation had a PRONOUNCED effect on the behavioral phenotype throughout the lifespan irrespective of sex”...

This whole paragraph with the different correlations makes no sense at all.

The results in females are very minor and why were estradiol levels not actually measured in these females? Did they also control estrous cyclicity in the treated females?

Also, the statement regarding kisspeptin having a protective effect in females (line 288) makes no sense at all.

In order to draw all these conclusions on this particular role of kisspeptin in the development of social and sexual behavior, additional work is needed to determine whether this is really directly related to kisspeptin and not through its general role on GnRH neuronal activity and the functioning of the HPG axis.

Version 1:

Reviewer comments:

Reviewer #1

(Remarks to the Author)

The revised manuscript can be considered significantly improved due to the addition of more samples, new experimental results quantifying HPG axis-related hormones in adults, revisions to the experimental data originally presented in Figure 6, and additional discussion in the Discussion section. However, I still have some concerns, particularly regarding the interpretation of these additional experiments.

Major Concerns:

· Page 7, Lines 236–257:

I am still unclear about the interpretation of the results in this section. In males, administration of Kp234 significantly elevated GnRH concentrations, while testosterone (T) and luteinizing hormone (LH) levels showed a trend toward increase, although not statistically significant. In contrast, in females, T and LH levels tended to decrease, while GnRH levels remained unchanged. Follicle-stimulating hormone (FSH) levels were unaffected in both sexes.

Taken together with the other results indicating Kp234 effects—including changes in T concentrations in neonatal pups, alterations in ultrasonic vocalization (USV) frequency in juveniles, and enhanced motivation for sexual behaviors in adults—these effects appear more pronounced in males than in females. Therefore, I suggest that the results shown in Figure 6 be interpreted more directly within this context, as mentioned above.

· Page 10, Lines 399–407:

In adults, there are substantial sex differences in the regulation of the HPG axis. Therefore, I believe that male and female data should be analyzed separately and discussed accordingly.

Additionally, as noted in Reviewer 2's comment, estradiol (E2) plays a more prominent role in regulating the HPG axis, especially in adult females, and testosterone is converted to E2 via aromatase. Thus, I believe it is critical to examine E2 levels as well.

The authors responded in the rebuttal, "...However, we recognize the challenge in extracting specific information about the estrous cycle from these data, compared to cytologic smears." However, the difficulty of determining the estrous cycle via vaginal smears and collecting samples at each stage for E2 measurement remains unclear.

While I believe that implementing this approach would improve the quality of the data, if it was not feasible, I recommend that the authors clearly explain the limitations or challenges in the Discussion section.

Minor Concerns:

- Graphical abstract:

I suggest that the authors differentiate the visual representation of the pronounced effects observed in males and the weaker effects in females, rather than using identical arrows (e.g., ↓ HPG Hormone Levels). This would help avoid overstating the results and contribute to a more accurate and balanced summary.

- Figure 2:

"Noenatal USVs (P6)" should be corrected to "Neonatal USVs (P6)".

Version 2:

Reviewer comments:

Reviewer #1

(Remarks to the Author)

While some questions remain for future investigation, the authors have sufficiently addressed the reviewer comments. I recommend acceptance of the manuscript.

Neonatal Kisspeptin Blockade Modulates Sex-Dependent Social Behaviors and Communication in Wistar Rats

Dear Editors and Reviewers,

We are grateful for the opportunity to revise our manuscript and are encouraged by your interest in its potential publication in the *Communications Biology Journal*.

We have carefully addressed the constructive and insightful comments provided by the reviewers, which have greatly contributed to improving the clarity, rigor, and overall quality of our work. Enclosed is the revised version of our manuscript with all changes highlighted in yellow, along with a detailed, point-by-point response to each of the reviewers' suggestions.

We appreciate the time and effort invested by the reviewers and editorial team, and we hope that the revisions meet your expectations.

Reply point-to-point to the comments of the Reviewers

Reply to Reviewer #1

Major concerns:

1 - Overall, the number of examples included in most experiments is insufficient. In particular, in the experiment in Figure 1, the number of samples is $n=3-4$, and in Figure 6, there is a group with only $n=2$. It is necessary to increase the number of samples to improve the validity of the results. Additionally, to validate the functions of neonatal kisspeptin, it will be more convincing to analyze whether the opposite effects of antagonist (Kp234) icv injection were found by kisspeptin agonist icv injection.

Reply: We appreciate the reviewer's concerns regarding the sample size in our experiments. We acknowledge that in the initial experiment assessing testosterone levels one hour after injection, the sample size was limited ($n=3-4$). To strengthen the results, we added new subjects from one additional group, which aligned with our previous data points. Although the sex imbalance in this group limited our ability to significantly increase the female sample size, we did not see a substantial change in significance. However, the additional female data maintained the trends observed in the previous results.

To address concerns about overall sample size, we have added a paragraph to the Methods section clarifying the calculation of our sample size, which was determined before the start of the experiments (lines 457-461): *"To avoid over-sampling and ensure adherence to the 3Rs principle, the sample size was calculated using GPower 3.1 (Universität Kiel, Kiel, Germany). The ANOVA: Fixed effects, special, main effects, and interactions was used, with an estimated effect size of 0.6, an α error probability of 0.05, a power of 0.95, a numerator df of 3, and 4 groups, resulting in a total N of 59, or approximately 14 per group."* Additionally, we have included partial eta-squared values for each significant statistical result

(calculation method detailed in the Statistical Analysis section, lines 588-590): “...to assess the effect size of the significant factor differences, partial eta-squared (η^2) was calculated as SS_{factor} (sum of squares of the factor) / SS_{total} (total sum of squares for all effects).” On average, our significant results yielded a partial eta-squared value of 0.23, which, according to Cohen (1998), can be converted into Cohen’s f ($\sqrt{\eta^2 / (1 - \eta^2)}$) and further to Cohen’s d ($2 * f * \sqrt{3 * (k-1) / (k+1)}$), where k is the number of means, 4), resulting in a Cohen’s d of 1.50. Based on Cohen’s guidelines (1988), this indicates a large effect size. Considering this statistical robustness and our commitment to animal welfare principles of reduction and refinement, we believe our sample sizes are adequate to support our hypotheses.

Additionally, we acknowledge the suggestion to investigate the effects of a kisspeptin agonist via ICV injection to validate the role of neonatal kisspeptin. However, since kisspeptin levels are already elevated at this time point, we did not expect to observe an opposite effect to the antagonism, as it would likely resemble the vehicle group. Nevertheless, this is a valuable suggestion that we will consider for future studies to gain a more comprehensive understanding of kisspeptin’s role in this context.

2 - In some experiments, for comparison between control vehicle (Veh) groups and Kp234 groups, statistical analyses were performed using males and females separately, while in others, using combined both sexes. However, since this work focuses on sexual dimorphism, the number of samples of both sexes should be increased and all analyses should be performed separately for males and females.

Reply: We appreciate the reviewer’s thoughtful critique regarding the statistical approach, which we clarified already and interpretation of our correlation analyses.

Our rationale was to combine certain groups due to (1) the limited sample size for a robust correlation analysis and (2) the fact that a two-way ANOVA revealed a significant treatment effect without a sex effect, indicating that treatment was the primary driver of changes in our data. However, we ensured that sex-specific differences were maintained where they were relevant.

Nonetheless, we explored the reviewer suggestion in Figure 3. We separated the correlation analyses for males and females and observed that vehicle-injected animals of both sexes exhibited significant correlations between Average Social Play and the number of USVs (Figure 3E), while antagonist-treated

groups did not. Interestingly, when further segregating by sex in the correlation between Average Social Play and Highest Frequency (Figure 3F), only males maintained a significant correlation. We recognize that this segregation added valuable depth to our findings, and we have revised the results section (lines 185-192): *“Additionally, we were interested in understanding if, on one hand, the USVs emitted during this social interaction were correlated with the social play activity, and on the other hand, the characteristics of those USVs influenced social behavior. Interestingly, only in the Veh groups, a significant Spearman correlation between Average Social Play and the number of USVs was observed in males vehicle (Fig 3E; $r=0.971$, $p\text{-value}=0.0111$) and females vehicle (Fig 3E; $r=0.6122$, $p\text{-value}=0.0261$). Regarding the correlation between the Average Play and the Highest Frequency only males vehicle presented a significant correlation (Fig 3F; $r=0.667$, $p\text{-value}=0.0414$).”* and discussion (lines 321-324): *“Notably, we found important USV disconnections between the number of calls and social behaviors in Kp234-injected groups, and more importantly, we uncovered a negative correlation between the mean high frequency and the average social play specifically in males.”* accordingly.

Additionally, after careful consideration, we have decided to remove the correlation analyses from Figure 6. Given the sample size limitations and the challenges in interpreting developmental-stage correlations, these analyses introduced more variability than clarity. By streamlining our data presentation, we aim to ensure that our findings remain focused, robust, and well-supported.

We appreciate the reviewer’s comments, which have helped us refine our manuscript to better reflect the significance of our findings while maintaining clarity and rigor in our statistical analyses.

3 - Page 4, Lines 139-141: Data in Figure 1 show that, by Kp234 icv administration, serum testosterone levels were significantly decreased in males, while increased in females at P6 stage. This suggests that neonatal kisspeptin blockage by Kp234 has inhibitory effects on masculinization or defeminization in males while it has excitatory effects in females. What brought about this sex difference? The authors should discuss some possible hypothesis or scheme. Moreover, if there is a long-lasting kisspeptin effect on development of neuronal circuits involved in social behaviors, this result seems inconsistent with the results showing shorter latency of sexual behaviors in Kp234-injected adult males. Because Kp234 inhibited masculinization of males at neonatal stage (P6), while it increased male's motivation for sexual behavior at mature adult stage (P60). The authors need to discuss this point as well.

Reply: We appreciate the reviewer's insightful comments regarding the sex differences observed in the effects of Kp234 administration. To clarify, the data presented in Figure 1 reflect testosterone levels measured one-hour post-injection (P0), not at P6. The primary conclusion drawn from this data is that our ICV injection effectively reduced the Neonatal Testosterone Peak in males.

In response to the reviewer's request for a discussion on the possible mechanisms underlying the observed sex differences, we have revised our discussion to delve deeper into the effects of Kisspeptin antagonism during the neonatal period. We hypothesize that the differential impacts of Kp234 on males and females may be attributed to distinct neuroendocrine pathways influenced by neonatal kisspeptin signaling, which could affect the development of relevant neuronal circuits.

Additionally, we recognize the potential inconsistency between the observed inhibition of masculinization at P0 and the increased motivation for sexual behavior in Kp234-treated males at P60. To address this point, we have highlighted in the discussion how the neonatal injection may have altered the latency to engage in sexual behaviors without fundamentally disrupting sexual performance (line 361-372): *"Additionally, while it may appear contradictory that neonatal Kiss blockade reduces sex-related behaviors yet is associated with increased sexual motivation in adulthood, we propose that these seemingly inconsistent findings reflect the engagement of distinct neural circuits. Specifically, we hypothesize those pre-copulatory behaviors-which are more closely linked to motivation and impulsivity-are governed by different pathways than those controlling consummatory sexual behaviors (such as mounting and intromission) in both males and females⁶¹. Thus, the neonatal injection may have altered the latency to engage in sexual behaviors by modulating circuits related to motivational drive, without fundamentally disrupting the underlying mechanisms of sexual performance. This interpretation suggests a differentiated modulation of neuronal circuitry, where the transient neonatal Kiss blockade impacts various aspects of sexual behavior through separate, functionally distinct networks."* We propose that these changes could be linked to alterations in other neural circuits related to motivation or impulsivity, rather than a direct impact on sexual differentiation. Thank you for highlighting these important aspects. We believe these additions enhance the clarity and depth of our discussion.

4 - Page 5, Lines 166-167: Data in Figure 2 show that Kp234 induced USVs with lower frequencies at neonatal stage (P6). It is important to examine whether this change in USV frequency functionally affects social interaction (with their mothers?). Additional analysis of maternal behaviors such as pup retrieval or pup care should be performed using Kp234 injected pups, vehicle injected pups, and their mothers.

Reply: We appreciate the reviewer's suggestion regarding the examination of the impact of changes in USV frequency on maternal behaviors, such as pup retrieval and care. We recognize the importance of

exploring the neonatal USV changes in the context of the maternal relationship. However, it is essential to clarify that the USVs measured in our study were induced by isolating the pups, which may not fully reflect their social interactions with their mothers. While we acknowledge that our group has previously investigated the correlation between USV impairment and maternal behaviors, we also noted that such studies often provide more insight into maternal behavior than the pup's social capacity (Ferreira et al., 2025, submitted and <https://hdl.handle.net/10316/106412>).

While we find this line of inquiry intriguing and agree it could yield valuable insights, we believe that examining maternal behaviors in conjunction with our current findings exceeds the scope of the present study. We will consider this suggestion for future research to further investigate the nuances of maternal-pup interactions and their implications.

5 - Page 5, Lines 173-174: Data in Figure 2 also show that, at juvenile stage (P29), Kp234 administration tended to increase USV frequency in males and conversely decrease in females and it is understandable that there was some difference between the sexes in Kp234 group. However, there is no significant difference between the Veh and Kp234 groups in both males and females. Although the authors argue that the effect of Kp234 is also present in P29, it is more reasonable to interpret these results as suggesting that Kp234 has little or no effect on USV frequency changes at P29.

Reply: We appreciate the reviewer's insightful comments on the interpretation of USV frequency data at P29. Although we did not observe any statistically significant differences at this stage, clear trends were evident across all parameters, especially in the high-frequency USVs (Figure 2H).

It is important to emphasize that the interaction effect between treatment (Vehicle vs. Kp234) and sex (male vs. female) suggests that both factors contributed to group differences in opposing directions. This finding underscores the complexity of the data and indicates that while the other parameters did not reach conventional significance, they still highlight meaningful trends.

We also recognize the challenges associated with sample size in this analysis, as we had to group animals due to technical constraints. Despite these challenges, the consistent trends observed in the USV frequency modulation point to an intriguing effect of Kp234 that warrants further investigation, even if not all differences achieved statistical significance. Importantly, looking at the effect size of this comparisons we found them to be of relatively high relevance despite their lack of significance (Principal frequency interaction: $\eta_p^2=0.19$; Peak frequency interaction: $\eta_p^2=0.19$; low frequency: $\eta_p^2=0.12$).

6 - Page 5, Lines 181-185: Data in Figure 3 show that Kp234 decreased juvenile social play behaviors at P29 by comparison of Veh group and Kp234 group. Is there significant difference when it is separately tested for males and females? As the authors argue that Kp234 effect on USV frequency in P29 is different between the sexes and that there is a correlation between USV and social play, it should be tested separately for males and females.

Reply: We appreciate the reviewer's concern regarding the separate analysis of males and females. In this study, we chose to present the p-values from the ANOVA main effects rather than focusing solely on post-hoc analyses, as the latter can sometimes be over-interpreted and may not fully capture the broader patterns in the data. While we recognize that this approach may be less conventional and, as the reviewer notes, potentially confusing, we believe it offers a more comprehensive view of the statistical results and helps mitigate the risk of overinterpretation.

Specifically, for the overall play behavior measure, we conducted a two-way ANOVA, which enabled comparisons across all groups while accounting for both sex and treatment. Although post-hoc tests did not reveal significant differences when males and females were analyzed separately, the ANOVA detected a significant treatment effect independent of sex. Thus, while we did examine sex-specific effects, these analyses did not reveal additional significant findings beyond those identified by the overall treatment effect.

7 -Pages 5-6, Lines 188-195: Similar to above, data in Figure 3 also show, only in Veh group, a positive correlation between the number of USVs and the number of social plays and a negative correlation between high frequency of USV and the number of social plays. This suggests that the individuals with higher number and lower frequency of USVs exhibit higher number of social plays. However, Kp234 appears to have almost no effect on USV frequency (Figure 2) or the number of USV (Figure S2) at P29. Nevertheless, Kp234 reduced the number of social plays. These results also seem inconsistent and this point should be addressed by the authors as well.

Reply: We appreciate the reviewer's observations regarding the relationship between USVs and social play behavior. The correlation observed in the Vehicle group suggests a natural link between vocalization and social interaction. However, as the reviewer noted, Kp234 administration reduced social play without significantly affecting USV number, but with minor effects on the frequency at P29, leading to an apparent inconsistency.

We propose that while USVs serve a social function under normal conditions, their role may be disrupted following neonatal Kisspeptin blockade. Our data suggest that Kp234 does not directly impact the number but influences the frequency of the calls, affecting their social relevance, as indicated by the lost correlation between USVs and social play in the Kp234 group. This aligns with our hypothesis that transient Kisspeptin blockade may alter the neuronal circuitry underlying social communication, particularly the main olfactory bulb-anterior olfactory nucleus-medial amygdala-hypothalamus pathway, as discussed in our manuscript.

Furthermore, while the correlation in the Vehicle group is significant, it does not imply causation. The mechanism by which Kp234 reduces social play may be independent of its effects on USVs, potentially involving altered motivation or sensory processing.

However, as mentioned before in this document we have followed the reviewer concern and separated the sexes on the correlation and performed the respective alterations to the document. By doing this, we hope to improve the clarity of the data and thank the reviewer for this suggestion.

8 - Page 6, Lines 208-210: Data in Figure 4 show that Kp234 did not affect anxiety-like behaviors but it increased total distance and mean velocity in open field test. Regarding this point, the authors argue that Kp234 can increase exploratory activity. It is questionable whether an increase in travel distance and velocity is equivalent to an increase in exploratory behavior. In fact, the result of open field test showed that there was no significant difference in "Time in center" and this may suggest that Kp234 affects not exploratory activity but locomotion itself. Additional behavior experiment such as Y-maze test is necessary to examine possible changes in exploratory activities. It should be noted that the possibility of Kp234 having some psychiatric effects such as excitement or hyperactivation cannot be ruled out. Such effects could potentially result in increased travel distance and velocity.

Reply: We appreciate the reviewer’s insights regarding the interpretation of the open-field test results. Although we initially suggested that increased total distance and velocity might indicate enhanced exploratory behavior, we acknowledge that these measures alone do not capture exploratory tendencies fully. Upon re-examination, we found that unsupported rearing—an activity linked to exploratory and locomotor behavior (see ref 38: <https://doi.org/10.1080/10253890.2018.1438405>)—exhibited a treatment effect, and we have now included these findings in the results (line 207-208): “...*unsupported rearing* ($F_{1,35}=5.5$, $p\text{-value}=0.0243$, $\eta_p^2=0.13$)”. This additional data further supports our hypothesis. Furthermore, as discussed, the observed increase in movement, combined with the reduced latency to engage in sexual behaviors, suggests a heightened environmental interest rather than mere hyperactivity. Regarding the suggestion of a Y-maze test, while this would provide additional insight into exploratory behavior and cognitive function, it falls outside the current study’s focus on early-life Kisspeptin blockade and its effects on social behaviors. We acknowledge the value of this approach and have added a note in the discussion (line 422): “...*such as Y-maze to access cognition.*” indicating that future studies should incorporate additional behavioral tests, such as the Y-maze or novel object recognition, to further elucidate the nature of Kp234’s effects. Lastly, we agree that the possibility of Kp234 influencing psychiatric-like behaviors (e.g., hyperactivity) warrants further investigation in dedicated studies.

9 - Page 6, Lines 224-226: Similar to above, data in Figure 5 show significantly shorter latency of sexual behaviors in adult males but no effect on female behaviors. At P6, Kp234 decreased testosterone levels in males and inhibited masculinization and probably development of neuronal circuits that are involved in male-type behaviors. Nevertheless, at P60, Kp234 males exhibited higher motivation for mating. I think this is also contradictory. Besides, it should be given a plausible explanation that testosterone levels in females increased at P6 by Kp234 (defeminization), but no effect on sexual behaviors in adults.

Reply: We appreciate the reviewer’s observations regarding the potential contradiction in our findings and welcome the opportunity to clarify. First, Figure 1 reports testosterone levels one hour after injection at P0—not P6. The main conclusion from this data is that our ICV injection effectively induced a transient reduction in the neonatal testosterone peak in males. However, we do not claim that this temporary reduction completely abolished the processes underlying masculinization.

Regarding sexual behavior, we maintain that the transient Kisspeptin blockade did not disrupt overall sexual function. Both males and females were able to reproduce naturally without notable behavioral impairments. The observed decrease in latency to initiate sexual behavior in males may instead reflect alterations in other neural circuits—such as those governing impulsivity or motivation—rather than a disruption in sexual differentiation itself. As for female behavior, our study specifically focused on overt sexual behavior. We acknowledge that additional behavioral paradigms would be valuable for exploring aspects such as sexual motivation in females. To address this, we have added a new paragraph in the Discussion (lines 361–378): *“Additionally, while it may appear contradictory that neonatal Kiss blockade reduces sex-related behaviors yet is associated with increased sexual motivation in adulthood, we propose that these seemingly inconsistent findings reflect the engagement of distinct neural circuits. Specifically, we hypothesize those pre-copulatory behaviors—which are more closely linked to motivation and impulsivity—are governed by different pathways than those controlling consummatory sexual behaviors (such as mounting and intromission) in both males and females⁶¹. Thus, the neonatal injection may have altered the latency to engage in sexual behaviors by modulating circuits related to motivational drive, without fundamentally disrupting the underlying mechanisms of sexual performance. This interpretation suggests a differentiated modulation of neuronal circuitry, where the transient neonatal Kiss blockade impacts various aspects of sexual behavior through separate, functionally distinct networks. Therefore, more studies are required to better understand how Kiss in this short and precise developmental stage modulates the neuronal networks that govern such behaviors. Also, we did not find differences in female sexual behavior, we hypothesize is due to the metrics we looked at being more related with consummatory behaviors rather than motivation and impulse, we recognize that new behavioral paradigms should be conducted in the future to access female sexual motivation, such as using bileveled chambers or odor preference. that elaborates on this hypothesis and interprets the observed results in that context.”*

Finally, we do not have confidence in that our intervention caused a substantial disruption of masculinization or feminization processes. Given the brief nature of the blockade and its confinement to a narrow neonatal window, we expect that key physiological mechanisms governing sexual differentiation—especially those activated during later stages such as puberty—remain largely intact. Nevertheless, we agree that further studies are warranted to elucidate the specific neural pathways through which neonatal Kisspeptin signaling may influence long-term sexual motivation and behavior.

10 - Page 7, Lines 234-236: As for the data shown in Figure 6 (A), I have several concerns. First, which two groups were tested resulting in $p=0.0008$? The authors argue that Kp234 affects kisspeptin receptor (KissR) expression. However, in male (also female), there was no significant difference between Veh and Kp234 group, suggesting that Kp234 does not affect the expression level of KissR. If there is KissR reduction by Kp234 in male adults, KissR reduction should cause impairment of functions of the HPG axis regulation. In turn, this impairment should negatively affect male’s sexual behavior, which is actually upregulated (Figure 5). This is also contradictory and needs to be addressed.

By comparing Veh groups of male and female, there is a significant sexual dimorphism in KissR expression (male>>female). Is this difference physiologically normal? Is there previous studies showing sex difference in KissR expression levels in rodents?

Reply: We appreciate the reviewer's questions regarding the data shown in Figure 6 (A). To clarify, the $p=0.0008$ refers to the significant effect of sex on Kiss1R expression, indicating that there is a sexual dimorphism in Kiss1R levels in the hypothalamus, with males exhibiting higher expression than females. Importantly, there was neither a treatment effect nor post-hoc significance between the Vehicle and Kp234 groups regarding hypothalamic Kiss1R levels in both males and females. This analysis was performed on overall hypothalamic samples, and while our results did not demonstrate significant differences, it does not imply that the expression levels are equal.

There is existing literature demonstrating sex differences in Kiss1R expression, which appear to vary depending on the specific hypothalamic nuclei examined. We believe that the tendency we observed in males could have reached significance if we had been able to analyze specific hypothalamic nuclei separately. To our knowledge, there are no previous reports documenting overall Kiss1R levels in the hypothalamus across sexes, therefore, we have in our lab a master's thesis student looking into it and hope to have new data soon, regarding this topic.

Regarding sexual behavior, it is important to note that the observed latency to engage in sexual behaviors may be influenced by motivational circuitry that does not directly involve the hypothalamus, has discussed in previous replies, and therefore, these results could coexist in the same hypothesis.

11 - Page 7, Lines 236-244: Data in Figure 6 (B) show that, in males, Kp234 significantly increased testosterone levels at P60 (adult stage), while tended to decrease them in females. By comparing Veh males and Veh females, it is suggested that testosterone levels are not different between sexually mature males and females. This result is not convincing because testosterone levels in mature males are normally much higher than those of mature females.

At P6 (neonatal stage), Kp234 decreased testosterone levels in males. In contrast, at P60 (adult stage), testosterone levels were higher in Kp234 male group than Veh male group. If early kisspeptin blockage caused inhibiting effect on masculinization, these result in P60 is also inconsistent with those of P6. The authors should discuss this point as well.

Reply: We appreciate the reviewer's observations regarding the testosterone levels presented in Figure 6 (B). To clarify, we do not suggest that a transient kisspeptin blockade in the neonatal period completely disrupts masculinization processes. Rather, we hypothesize that this critical period may primarily influence social behavior, with potential effects on other aspects of development as well. It is important to note that while our analysis focused on the hypothalamus and the HPG axis, we cannot disregard possible changes in other brain regions that may impact behavioral outcomes, as we discussed in the manuscript (line 411-414): *"However, it is important to remember the widespread nature of the neonatal blockade that could have influenced other mechanisms in other brain regions and, therefore, contributed to the observed phenotype."*

We found the lack of significant differences between testosterone levels in Vehicle males and females intriguing. To further investigate this, we rerun the samples in a new testosterone kit and performed an ELISA for GnRH, LH, and FSH levels, which are also included in Figure 6. This time we found the previsible sex difference in testosterone levels in the Vehicle group. Furthermore, this additional analysis confirmed

our findings and revealed that GnRH and LH levels exhibited similar trends to testosterone levels, which is in accordance with the HPG axis function. We did not find differences in the FSH and the ratio FSH/LH which is described in humans and rodents to be an important metric to assess good physiological reproductive health. These results strengthen our previous results. We have changed the results with this new data (line 236-255): *“The levels of serum GnRH revealed a significant effect of sex ($F_{1,32}=6.00$, p -value= 0.0200, $\eta^2=0.13$), treatment ($F_{1,32}=4.30$, p -value= 0.0462, $\eta^2=0.09$), and an interaction effect ($F_{1,32}=5.07$, p -value= 0.0313, $\eta^2=0.11$). These changes were mainly due to the significant difference ($q=4.809$, p -value=0.0094) between males Kp234 (5.385 ± 2.241 pg/ml) and females Kp234 (5.240 ± 2.788 ng/ml), and additionally the increase ($q=4.233$, p -value=0.0258) between males Veh (6.347 ± 2.710 pg/ml) and Kp234 (5.385 ± 2.241 pg/ml). LH levels presented the same trends, with a significant sex effect ($F_{1,33}=8.3$, p -value= 0.0067, $\eta^2=0.18$) and interaction effect ($F_{1,33}=5.7$, p -value= 0.0226, $\eta^2=0.12$). Again, motivated by a difference ($q=5.339$, p -value=0.0034) between males (2626 ± 617 ng/ml) and females (1696 ± 336.7 ng/ml) injected with Kp234. Finally, testosterone levels followed the same trend having a significant main effect of sex ($F_{1,30}=12.35$, p -value= 0.0014, $\eta^2=0.24$) and an interaction effect ($F=5,841$, p -value= 0.0219, $\eta^2=0.12$). These changes were mainly due to the significant difference ($q=6.557$, p -value=0.0004) between males Kp234 (2097.654 ± 763.620 ng/ml) and females Kp234 (848.303 ± 671.320 ng/ml), and additionally the increase ($q=3.858$, p -value=0.0490) between males Veh (1302.903 ± 393.131 ng/ml) and Kp234 (2097.654 ± 763.620 ng/ml). Then we looked at the levels of FSH and the ratio LH/FSH which is described in the literature as predictors of reproductive function³⁹, but we found no differences. Finally, we found a significant main effect of sex ($F_{1,17}=16.78$, p -value=0.008, $\eta^2=0.44$) on protein levels of KissR (Fig 6F).” and have discussed the in detail in the discussion section (Line 389-399): *“Additionally, serum levels of gonadotropin-releasing hormone (GnRH), luteinizing hormone (LH), and testosterone exhibited patterns consistent with normal fluctuations of the HPG axis, where increased GnRH leads to elevated LH and subsequently higher testosterone levels⁶³. These results indicate that neonatal Kiss perturbation modifies hormonal regulation within the HPG axis in a sex-specific manner, as evidenced by significant interaction effects. Notably, no differences were observed in follicle-stimulating hormone (FSH) levels or the LH/FSH ratio, metrics associated with spermatogenesis and ovulatory regulation^{39,64}. This supports our hypothesis that neonatal Kiss signaling plays a more critical role in reprogramming hormonal systems and influencing specific behaviors, rather than directly affecting sexual performance or reproductive health.”.**

12 - Page 7, Lines 245-249: The authors argue that Kp234 can affect the feedback loop of the HPG axis regulation. However, these data showed only KissR and testosterone levels and, in order to confirm this Kp234 effect, it is necessary to examine expression levels of other key factors (hormones) of the HPG axis regulation, such as GnRH, LH, FSH, estrogen, and kisspeptin. Quantitative PCR (qPCR) will be the most effective analysis to examine the gene expression levels of these factors.

There is no correlation between KissR expression and testosterone levels in Veh groups, suggesting that KissR is expressed regardless of testosterone levels at P60. If a positive correlation between these in Kp234 group is correct, it is not consistent with the result that testosterone level was increased (B) but KissR expression is decreased (A) in Kp234 group. It is difficult to interpret these different correlation patterns between the two groups.

Reply: We appreciate the reviewer's suggestion regarding a more comprehensive analysis of key hormones involved in HPG axis regulation. In response to this concern, we have now measured GnRH, LH, and FSH serum levels using the same samples used for testosterone analysis and have incorporated these findings into Figure 6 (C-E).

Interestingly, we observed that GnRH and LH levels followed the same pattern as testosterone, further supporting the hypothesis that neonatal Kp234 administration induces long-term effects on the HPG axis. Regarding FSH levels, we have addressed these findings in more detail in the revised discussion (Line 389-399): "Additionally, serum levels of gonadotropin-releasing hormone (GnRH), luteinizing hormone (LH), and testosterone exhibited patterns consistent with normal fluctuations of the HPG axis, where increased GnRH leads to elevated LH and subsequently higher testosterone levels⁶³. These results indicate that

neonatal Kiss perturbation modifies hormonal regulation within the HPG axis in a sex-specific manner, as evidenced by significant interaction effects. Notably, no differences were observed in follicle-stimulating hormone (FSH) levels or the LH/FSH ratio, metrics associated with spermatogenesis and ovulatory regulation^{39,64}. This supports our hypothesis that neonatal Kiss signaling plays a more critical role in reprogramming hormonal systems and influencing specific behaviors, rather than directly affecting sexual performance or reproductive health.”. Briefly, in females, Kp234 treatment resulted in more variable FSH and LH levels, which may reflect a broader distribution across different stages of the estrous cycle rather than a direct effect of the antagonist. In males, we observed a tendency for increased LH levels while FSH remained comparable to vehicle controls. Given that LH primarily regulates testosterone production in males while FSH is more involved in spermatogenesis, these findings reinforce the idea that neonatal Kp234 administration exerts lasting modulatory effects on the HPG axis without overtly disrupting reproductive function.

After careful consideration, we also acknowledge the reviewer’s concern regarding the difficulty in integrating the correlation analysis between Kiss1R expression and testosterone levels. To improve the clarity and focus of our findings, we have removed these correlations from Figure 6, as they did not contribute substantially to our main conclusions.

We appreciate the reviewer’s valuable feedback, which has helped refine our analysis and interpretation of the data.

13 - Page 7, Lines 252-266: Similar to above, the different correlations between these two groups are also difficult to interpret and the correlation heatmap in Fig. 6 (D) is very complicated and difficult to interpret. For example, how do the authors explain the result that there is a positive correlation between the frequency of USVs in juveniles (P29) and the expression levels of KissR in Adults (P60)? As they are separate indices at different developmental stages, it seems unlikely that these two indices have some clear relationship.

Reply: We appreciate the reviewer’s critical assessment of the correlation heatmap in Figure 6(D) and acknowledge the difficulty in interpreting these complex relationships, particularly across different developmental stages. After careful consideration, we agree that these correlations did not provide meaningful support for our main conclusions and could potentially introduce unnecessary complexity. Therefore, we have removed the correlation heatmap from Figure 6, as well as the corresponding supplementary table. Additionally, we have revised the discussion section to exclude references to these correlations, ensuring that our interpretation remains focused on the direct findings. By doing so, we aim to present our data in a clearer and more unbiased manner, allowing the results to stand on their own without overinterpreting associations that may not be biologically relevant.

14 - Pages 8, Lines 311-318: I believe that histological analysis using Kp234 icv injected rats to examine whether neonatal kisspeptin affects developmental regulation of neural circuits that are involved in social behaviors will be direct evidence for this part of the argument. Regarding classical neurotransmitters such as GABA, which is mentioned in Page9, line360, it would be very insightful to examine potential alterations in the number of neurons or projection sites.

Reply: We appreciate this insightful suggestion and agree that investigating the effects of neonatal Kisspeptin perturbation on the developmental regulation of neural circuits involved in social behaviors

would provide direct evidence to support our findings. Similarly, analyzing potential alterations in classical neurotransmitter systems, such as GABAergic signaling, through histological examination of neuron numbers or projection sites, could further elucidate the underlying mechanisms. However, these investigations extend beyond the scope of the current study. We acknowledge the importance of these questions and consider them valuable directions for future research.

15 - Pages 10, Lines 380-381: As with the above, it is not persuasive to make this argument because there was no significant difference between the sexes in the Veh group in Figure 6 (B).

Reply: We appreciate the reviewer's concern regarding this point. As mentioned previously, we rerun the samples and now show the expected sex difference in the Vehicle group.

Minor:

1 - Graphical abstract: Under the list with arrows indicating the actions of Kp234 icv, the text "Neonatal Kiss induces~" is presented, which makes it difficult to tell whether this list shows the action of Kp234 or the function of endogenous kisspeptin. It would be preferable to have a heading such as "Kiss antagonist effects" above the list.

Reply: We have made this change to improve clarity. The heading now explicitly states "Kiss antagonist effects" to clearly indicate that the listed actions refer to Kp234 icv administration.

2 - Supporting information S1 (C): It seems that there may be a significant difference in males between PND10-12. Is this really not significant (n.s.)?

Reply: We appreciate the reviewer's comment regarding the potential difference in males between PND10-12. However, as shown in our statistical analysis, while the three-way ANOVA revealed a significant effect of "Days × Treatment" ($p = 0.0030$), the specific post-hoc multiple comparisons did not yield any statistically significant differences between groups, including at PND10-12. Given this, we must conclude that while there may be a visible trend in the data, it does not reach statistical significance.

3 - Page 5, Line 163: it should be written as "q=4.968".

Reply: We appreciate the correction and have revised the text accordingly to reflect "q=4.968.", by adding an equal sign (line 160): "(Fig 2C; Veh: 58.821 ± 4.369 kHz; Kp234: 49.186 ± 5.531 kHz; q=4.968, p-value=0.0081)"

Reply to Reviewer #2

Reviewer #2 (Remarks to the Author):

1. The main issue with the paper is the over-interpretation of the data, contributing a direct role to kisspeptin to the development of social and sexual behavior. There is no discussion/mention of the role of kisspeptin in stimulating GnRH neurons and how all these effects might just simply be due to decreased GnRH signaling and decreased stimulation of gonadal sex hormones (which obviously was the case regarding testosterone levels), which has affected the process of masculinization/defeminization to some extent. There is also no data presented on the duration of these effects of blocking kisspeptin receptors. Did they actually compare different doses and timelines of their treatment? They just measured testosterone levels once, 1h after the icv treatment.

Reply: We appreciate the reviewer's insights regarding the potential over-interpretation of our findings related to kisspeptin and its influence on social and sexual behavior. We acknowledge the significance of discussing kisspeptin's role in stimulating GnRH neurons and how alterations in GnRH signaling may contribute to observed changes in gonadal hormones, particularly testosterone levels.

To address this, we conducted an ELISA analysis of GnRH levels in adulthood (Figure 6A), revealing that GnRH levels followed similar trends to testosterone. This finding supports the notion that the effects we observed on testosterone arise from GnRH over-stimulation. However, we must also recognize that the widespread distribution of kisspeptin receptors throughout the brain suggests that the consequences of blocking these receptors may extend beyond the HPG axis, potentially affecting regions crucial for social behavior, such as the amygdala, as we discussed already (line 411-414): *"However, it is important to remember the widespread nature of the neonatal blockade that could have influenced other mechanisms in other brain regions and, therefore, contributed to the observed phenotype."* . While we cannot discount the influence of the HPG axis, it is essential to consider potential effects from other brain regions, and this line of questioning deserves a follow-up in the future.

This article aims to highlight the importance of kisspeptin in social behavior, and thus did not delve deeply into the underlying mechanisms, which we acknowledge warrant further investigation.

Regarding the duration of kisspeptin receptor blockade effects, we understand the importance of assessing various doses and treatment timelines. Although we measured testosterone levels only once, one-hour post-injection, this time point was chosen based on prior literature, including the study by Roseweir et al. (2009), which guided our selection of antagonist concentration and timing of sacrifice, for better comprehension we added a phrase regarding this issue on the method section (line 479-480): *"The timepoint, antagonist selection and concentration was chosen based on Roseweir et al. (2009)⁶⁸."* .

2. The paper is just filled with overstatements, e.g. line 250, "since temporary Kiss pathway modulation had a PRONOUNCED effect on the behavioral phenotype throughout the lifespan irrespective of sex"... This whole paragraph with the different correlations makes no sense at all.

Reply: We appreciate the reviewer's feedback regarding the statements made in our manuscript, particularly concerning line 250 and the interpretation of the correlation heatmap in Figure 6(D). We understand the concern that certain phrases, such as "temporary Kiss pathway modulation had a

PRONOUNCED effect on the behavioral phenotype throughout the lifespan irrespective of sex,” may suggest overstatements.

We agree that the correlations previously presented did not provide robust support for our conclusions and could lead to confusion. Consequently, we have removed the correlation heatmap from Figure 6, along with the corresponding supplementary table. Furthermore, we have revised the discussion section to eliminate references to these correlations, allowing us to focus on the direct findings of our study.

By refining our language and ensuring that our interpretation is grounded in the observed data, we aim to present a clearer and more accurate narrative. We appreciate the reviewer’s insights, which have helped us strengthen our manuscript and improve the clarity of our conclusions regarding the effects of kisspeptin modulation on behavioral phenotypes.

3. The results in females are very minor and why were estradiol levels not actually measured in these females? Did they also control estrous cyclicity in the treated females?

Also, the statement regarding kisspeptin having a protective effect in females (line 288) makes no sense at all.

Reply: We appreciate the reviewer’s insightful comments regarding the results in females. We acknowledge that the findings in females were less pronounced compared to those in males. We measured LH and FSH levels in the treated females (Figure 6). However, we recognize the challenge in extracting specific information about the estrous cycle from these data, compared to cytologic smears. Our observations indicated that females treated with Kp234 exhibited more variability in their hormone levels compared to the vehicle group, which may suggest bigger distribution throughout the estrous cycle. We acknowledge this as a limitation of our study (line 404-405): *“Our study did not account for the female estrous cycle, which could contribute to observed variations.”*

We also recognize that the statement regarding kisspeptin having a protective effect in females, is a hypothesis that requires a more thoughtful investigation. We revised this statement to identify it has a hypothesis that requires more research and avoid any misinterpretation, (line 281-289). *“Given our findings, we propose that the observed sex differences may stem from distinct neuroendocrine pathways activated by neonatal Kiss signalling. In males, blocking Kiss reduces systemic testosterone levels, likely disrupting the normal neonatal testosterone surge that contributes to brain masculinization. In contrast, our findings seem to point for a tendency of females to have an increase in the testosterone levels upon antagonism of Kiss1r. This may imply, to an extent, that Kiss in this neonatal period serves different purposes depending on the sex. However, further research is warranted to delineate the precise mechanisms and neural circuits involved in these divergent effects.”*. We intended to highlight potential differences in the effects of kisspeptin modulation between sexes, and we will ensure this is articulated clearly in the revised manuscript.

4. In order to draw all these conclusions on this particular role of kisspeptin in the development of social and sexual behavior, additional work is needed to determine whether this is really directly related to kisspeptin and not through its general role on GnRH neuronal activity and the functioning of the HPG axis.

Reply: We appreciate the reviewer’s thoughtful critique regarding the interpretation of our findings on kisspeptin’s role in the development of social and sexual behavior. We acknowledge that establishing a direct causal link between kisspeptin and these behavioral outcomes requires further investigation,

particularly considering kisspeptin's established role in regulating GnRH neuronal activity and the hypothalamic-pituitary-gonadal (HPG) axis.

In response to this concern, we have expanded our analysis to include GnRH levels in adulthood, which were measured using ELISA (Figure 6A). Our results indicated that GnRH levels mirrored the trends observed with testosterone. However, we recognize the widespread distribution of kisspeptin receptors throughout the brain, suggesting that kisspeptin antagonism could influence regions critical to social behavior beyond the HPG axis, such as the amygdala.

While we do acknowledge the significance of the HPG axis in our findings, we believe that our study highlights the potential multifaceted role of kisspeptin in influencing behavioral outcomes. We agree that a comprehensive mechanistic analysis is warranted, and we emphasized in the revised manuscript that further research is necessary to disentangle the direct effects of kisspeptin from its broader regulatory role within the HPG axis (line 389-408): *"Additionally, serum levels of gonadotropin-releasing hormone (GnRH), luteinizing hormone (LH), and testosterone exhibited patterns consistent with normal fluctuations of the HPG axis, where increased GnRH leads to elevated LH and subsequently higher testosterone levels⁶³. These results indicate that neonatal Kiss perturbation modifies hormonal regulation within the HPG axis in a sex-specific manner, as evidenced by significant interaction effects. Notably, no differences were observed in follicle-stimulating hormone (FSH) levels or the LH/FSH ratio, metrics associated with spermatogenesis and ovulatory regulation^{39,64}. This supports our hypothesis that neonatal Kiss signaling plays a more critical role in reprogramming hormonal systems and influencing specific behaviors, rather than directly affecting sexual performance or reproductive health.*

Importantly, we observed a significant difference in testosterone levels between male and female vehicle-treated groups. Which is consistent with GnRH and LH levels. Importantly, existing literature indicates that testosterone levels can be influenced by factors such as circadian rhythms, sociability, and females also experience testosterone fluctuations ⁶⁵. Our study did not account for the female estrous cycle, which could contribute to observed variations. Recent research highlights the role of testosterone pulses in females⁶⁵, suggesting that our findings may reflect genuine physiological differences that warrant further investigation. We acknowledge that more targeted studies are necessary to fully elucidate these observations." This additional work will be crucial in solidifying our conclusions regarding kisspeptin's specific contributions to social and sexual behaviors.

Thank you for your insightful feedback, which will aid in refining our discussion and enhancing the clarity of our manuscript.

Neonatal Kisspeptin Blockade Modulates Sex-Dependent Social Behaviors and Communication in Wistar Rats

Dear Editors and Reviewers,

We are grateful for the opportunity to revise and resubmit our manuscript to Communications Biology. We appreciate the reviewers' insightful comments and the editorial team's support, which have been invaluable in refining our study.

We have carefully addressed all suggestions, with particular attention to the interpretation of the newly submitted data. These revisions have substantially improved the clarity and quality of our manuscript. The revised version includes all changes highlighted, and we also provide below a detailed, point-by-point response to the reviewers' comments.

We thank the reviewers and editors once again for their time and thoughtful feedback. We hope the revised manuscript meets your expectations and look forward to your evaluation.

Reply point-to-point to the comments of the Reviewers

Major concerns:

1 - I am still unclear about the interpretation of the results in this section (Page 7, Lines 236–257). In males, administration of Kp234 significantly elevated GnRH concentrations, while testosterone (T) and luteinizing hormone (LH) levels showed a trend toward increase, although not statistically significant. In contrast, in females, T and LH levels tended to decrease, while GnRH levels remained unchanged. Follicle-stimulating hormone (FSH) levels were unaffected in both sexes.

Taken together with the other results indicating Kp234 effects—including changes in T concentrations in neonatal pups, alterations in ultrasonic vocalization (USV) frequency in juveniles, and enhanced motivation for sexual behaviors in adults—these effects appear more pronounced in males than in females. Therefore, I suggest that the results shown in Figure 6 be interpreted more directly within this context, as mentioned above.

Reply: Thank you for this important observation. We have thoroughly revised the manuscript to clarify and better contextualize these results. Specifically, we highlight that while some early effects (e.g., P6 USV frequency) were observed across sexes, many outcomes demonstrated a sex-specific interaction, with effects being more pronounced in males.

To reflect this, we revised the Results and Discussion sections, in the following:

- Results (Line 135): "...reduced upon Kiss blockade, mainly affecting males."
- Results (Line 143-144): "...frequency, by accentuating the sexual dimorphism of USV emission in a social context..."

- Results (Line 230-232): "...Additionally, neonatal Kiss blockade appears to enhance hormonal sex differences while preserving reproductive function."
- Discussion (Line 238-239): "...by reducing the gap between males and females, mainly due to a loss of male-specific behavior."
- Discussion (Line 280-281): "...although males are more affected, kiss acts irrespective of sex in some contexts."
- Discussion (Line 376-377): "..., again, mainly explained by the increased hormone levels in males Kp234 which increased in this condition a more pronounce sex dimorphism, ..."

These revisions strengthen our argument that neonatal kisspeptin signaling contributes to long-term, sex-differentiated modulation of the HPG axis and social behaviors.

2- Page 10, Lines 399–407: In adults, there are substantial sex differences in the regulation of the HPG axis. Therefore, I believe that male and female data should be analyzed separately and discussed accordingly. Additionally, as noted in Reviewer 2's comment, estradiol (E2) plays a more prominent role in regulating the HPG axis, especially in adult females, and testosterone is converted to E2 via aromatase. Thus, I believe it is critical to examine E2 levels as well. The authors responded in the rebuttal, "...However, we recognize the challenge in extracting specific information about the estrous cycle from these data, compared to cytologic smears." However, the difficulty of determining the estrous cycle via vaginal smears and collecting samples at each stage for E2 measurement remains unclear. While I believe that implementing this approach would improve the quality of the data, if it was not feasible, I recommend that the authors clearly explain the limitations or challenges in the Discussion section.

Reply: We appreciate this insightful comment. While analyzing sexes separately (e.g., via t-tests) could yield clearer intra-sex effects, our central objective was to assess sex-specific modulation by neonatal kisspeptin blockade. Therefore, we chose statistical methods that emphasize interaction effects (e.g., two-way ANOVA), which directly reflect our hypothesis of sex-dependent impact.

We also agree on the crucial role of estradiol, particularly in the female HPG axis. Unfortunately, due to limitations in our experimental design, estradiol levels were not measured, nor did we monitor estrous cycle stages. We now acknowledge this limitation clearly in the Discussion (Line 389-393):

"Our study did not account for the female estrous cycle, nor we were able to measure estradiol, which is the active form of testosterone in the brain by the action of the aromatase, which could contribute to observed variations. Future studies should consider the effect a neonatal temporary Kiss blockade plays in the activity of aromatase, by measuring estradiol and receptor levels later in life."

We thank the reviewer for encouraging us to expand this critical discussion.

Minor:

1 - Graphical abstract:I suggest that the authors differentiate the visual representation of the pronounced effects observed in males and the weaker effects in females, rather than using identical arrows (e.g., ↓ HPG Hormone Levels). This would help avoid overstating the results and contribute to a more accurate and balanced summary.

Reply: We revised the graphical abstract accordingly. Male results are now presented in bold with enlarged symbols, while female effects are displayed with lighter arrows, more accurately reflecting the strength of observed differences.

2 - "Noenatal USVs (P6)" should be corrected to "Neonatal USVs (P6)".

Reply: We appreciate the correction and have edited figure 2.